# Metabolic phenotyping of BMI to characterize cardiometabolic risk: evidence from large population-based cohorts

Habtamu B. Beyene [1,2,3,4,12], Corey Giles [1,3,4,12], Kevin Huynh [1,3,4], Tingting Wang[1,3,4], Michelle Cinel[1], Natalie A. Mellett[1], Gavriel Olshansky [1], Thomas G. Meikle[1,3], Gerald F. Watts [5,6], Joseph Hung [5], Jennie Hui[7,8,9], Gemma Cadby [9], John Beilby[8], John Blangero [10], Eric K. Moses [8,11], Jonathan E. Shaw [1,2], Dianna J. Magliano [1,2,13] ✉ & Peter J. Meikle [1,2,3,4,13] ✉

Obesity is a risk factor for type 2 diabetes and cardiovascular disease. However, a substantial proportion of patients with these conditions have a seemingly normal body mass index (BMI). Conversely, not all obese individuals present with metabolic disorders giving rise to the concept of "metabolically healthy obese". We use lipidomic-based models for BMI to calculate a metabolic BMI score (mBMI) as a measure of metabolic dysregulation associated with obesity. Using the difference between mBMI and BMI (mBMIΔ), we identify individuals with a similar BMI but differing in their metabolic health and disease risk profiles. Exercise and diet associate with mBMIΔ suggesting the ability to modify mBMI with lifestyle intervention. Our findings show that, the mBMI score captures information on metabolic dysregulation that is independent of the measured BMI and so provides an opportunity to assess metabolic health to identify "at risk" individuals for targeted intervention and monitoring.

The prevalence of obesity and overweight is growing worldwide[1,2]. According to recent estimates, some 30% of men and 35% of women are obese in many countries including in North America, the Middle East, Asia, and Australia[3]. The progression of obesity is influenced by various factors such as age, gender, ethnicity, level of education, genetic predisposition, and lifestyle choices[4,5]. Excess body weight, which is a key characteristic of obesity, can be partially attributed to a combination of high calorie intake and insufficient physical exercise[6,7]. Consequently, adopting healthy eating habits (e.g., low carbohydrate intake)[8] and engaging in regular physical exercise have been consistently linked to reduced odds of obesity and central obesity[4,9]. Obesity is strongly associated with an increased risk of cardiometabolic disorders including type 2 diabetes mellitus (T2DM)[10,11] and cardiovascular disease (CVD)[12,13].

Body mass index (BMI), defined as weight divided by height squared (kg/m$^2$) is an accessible surrogate measure of obesity. Compared with direct measures of adiposity, such as computed tomography and dual energy x-ray absorptiometry, BMI is an inexpensive, simple and easily interpretable metric. World Health Organization (WHO) provides classifications and standardized cut-off points.

[1]Baker Heart and Diabetes Institute, Melbourne, VIC, Australia. [2]Faculty of Medicine, Nursing and Health Sciences, Monash University, Melbourne, VIC, Australia. [3]Baker Department of Cardiovascular Research Translation and Implementation, La Trobe University, Melbourne, VIC, Australia. [4]Baker Department of Cardiometabolic Health, Melbourne University, Melbourne, VIC, Australia. [5]School of Medicine, University of Western Australia, Perth, WA, Australia. [6]Lipid Disorders Clinic, Department of Cardiology, Royal Perth Hospital, Perth, WA, Australia. [7]PathWest Laboratory Medicine of Western Australia, Nedlands, WA, Australia. [8]School of Biomedical Sciences, University of Western Australia, Crawley, WA, Australia. [9]School of Population and Global Health, University of Western Australia, Crawley, WA, Australia. [10]South Texas Diabetes and Obesity Institute, The University of Texas Rio Grande Valley, Brownsville, TX, USA. [11]Menzies Institute for Medical Research, University of Tasmania, Hobart, TAS, Australia. [12]These authors contributed equally: Habtamu B. Beyene, Corey Giles. [13]These authors jointly supervised this work: Dianna J. Magliano, Peter J. Meikle. ✉e-mail: Dianna.Magliano@baker.edu.au; peter.meikle@baker.edu.au

Specifically, an individual whose BMI falls between 18.5 and 24.9 is considered a normal weight; 25.0−29.9, overweight; and 30.0 or higher representing obese. Despite not directly measuring body composition and adiposity, BMI strongly associates with cardiometabolic outcomes[14]. However, it has been recognized that not all individuals who are obese/overweight−based on measured BMI−present with an increased risk of metabolic complications[15]. A specific group of individuals who are obese, but "metabolically healthy", have been reported in multiple population cohort studies[16,17]. Conversely, certain individuals, whom are within normal BMI range, are metabolically unhealthy, resulting in an increased risk for cardiometabolic disease[18,19].

Several studies have identified profound perturbations in circulating lipids associated with obesity[20–22]. In addition, we have previously shown that the plasma lipidome is strongly associated with BMI, with several hundred plasma lipid species significantly associated in large population cohorts[23,24]. Of note, positive associations of triacylglycerol, diacylglycerol, deoxyceramide and sphingomyelin, and negative associations of lysophosphatidylcholine and ether-lipid species have been consistently reported with BMI[23,25,26] highlighting the potential impact of obesity on multiple lipid metabolic pathways. In contrast to some genetic loci stringently associated with BMI which explain less than 3% of phenotypic variation of BMI[27], metabolism, driven by multiple environmental factors (diet, exercise and other exposures), can explain up to 49% of BMI variability[21,22]. Importantly, in several prospective studies, many BMI associated metabolites (including lipids) were also markedly associated with risk of diabetes[27–29] and CVD[30–32] independent of BMI. These findings convey an important message about the potential of metabolic phenotyping to refine the obesity definition beyond BMI measurements.

The strong associations of lipids and other metabolites with BMI has raised the prospect of developing metabolic scores that better capture the hidden risk of cardiometabolic diseases, i.e. the risk not explained by BMI itself, as in normal weight but metabolically unhealthy individuals. Using the human metabolome, Cirulli et al. identified metabolic signatures that distinguish healthy obese and normal weight individuals with abnormal metabolic profile[21]. Of note, individuals who were classified as obese based on their metabolome, had 2 to 5 times higher risk of cardiovascular events compared to their counterparts with similar BMI but opposing metabolic signature. Moreover, a recent study has showed that, lean individuals with abnormal metabolism related to obesity had higher risk of developing T2DM and all-cause mortality compared to those individuals with lean BMI and healthy metabolism[33]. The human lipidome has also been used to model BMI where it explained up to 47% of BMI variation with just 75 predictors in a LASSO model[22]. Moreover, a study by Watanabe et al.[34] had recently demonstrated the power of a multi-omics profiling in uncovering population heterogeneity within both health and disease states. The study further showed that, the metabolome inferred BMI was substantially decreased in response to lifestyle coaching compared to the actual BMI. Taken together, these findings suggest the potential utility of the human lipidome and or metabolome to characterizing the heterogeneity in obesity and identify individuals at an increased risk of obesity-related diseases.

These early studies have identified mBMI scores that capture residual risk of a range of cardiometabolic outcomes. However, the signal being captured by metabolic BMI scores has not been clearly defined nor has the relationship with disease outcomes been adequately quantified. To address this, we developed models to predict BMI and calculated mBMI scores using plasma lipidomic data in a large Australian cohort−the Australian Diabetes, Obesity and Lifestyle Study (AusDiab; $n = 10,339$) (Fig. 1a, b). Metabolic BMI scores were validated in an independent Australian cohort, the Busselton Health Study (BHS, $n = 4492$) (Fig. 1c). The mBMI score, and a derived score from the difference between mBMI and measured BMI (mBMIΔ), were

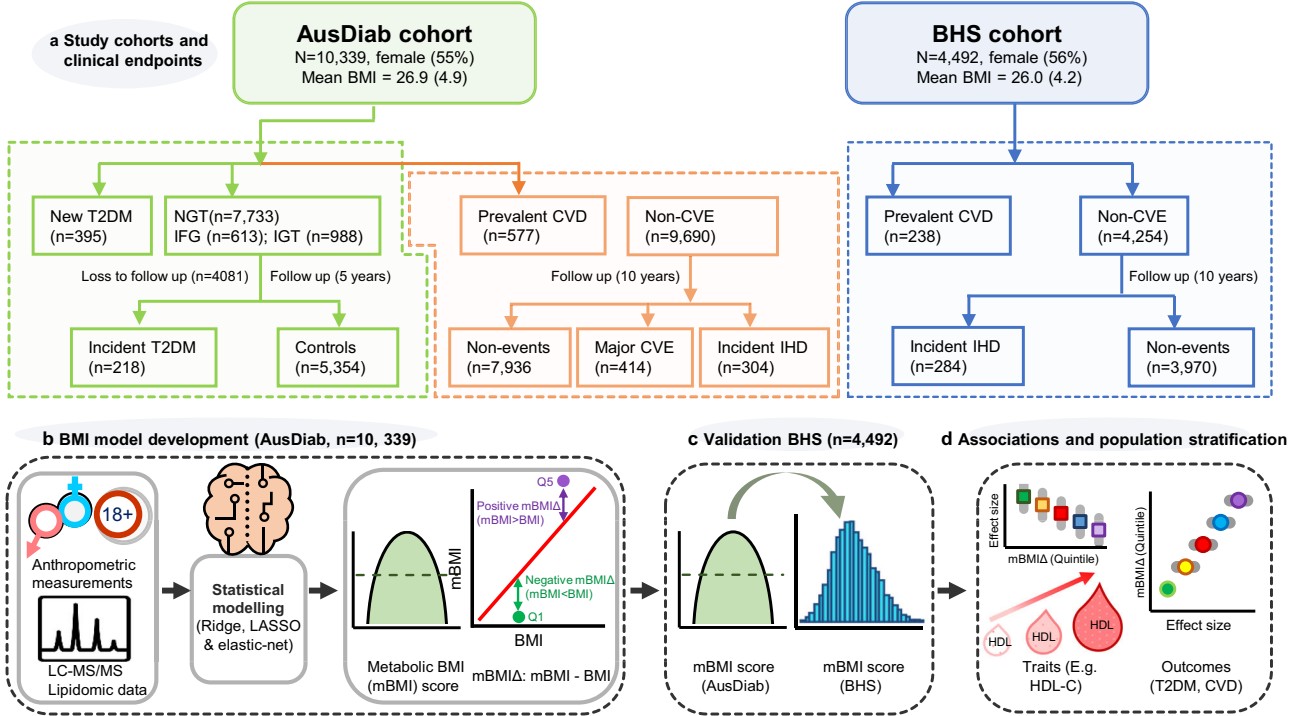

**Fig. 1 | An overview of the study design for the development of metabolic BMI scores and the subsequent downstream analyses. a** Study participants and clinical end-points in the AusDiab and BHS cohorts. **b** BMI model development: lipidomic data was used for the generation of the metabolic BMI score in the discovery cohort (AusDiab) using linear models. **c** External validation of the mBMI score in the BHS cohort. **d** Downstream analyses (association of the metabolic BMI scores with cardiometabolic traits and outcomes). AusDiab Australian Diabetes, Obesity and Lifestyle Study, BHS Busselton Health Study, BMI body mass index, mBMI metabolic BMI, mBMIΔ metabolic BMI delta, IGT impaired glucose tolerance, IFG impaired fasting glucose, NGT normal glucose tolerance, T2DM type 2 diabetes mellitus, CVD cardiovascular disease, CVE cardiovascular event, IHD ischemic heart disease, LC-MS/MS liquid chromatography tandem mass spectrometry.

examined for their association with metabolic traits, the lipids used to generate the scores and with prevalent-, and incident-cardiometabolic outcomes (Fig. 1d).

Here, we show that mBMIΔ captures a metabolic signal that is independent of BMI, but closely mirrors the BMI signal. This provides an independent measure of the metabolic dysregulation associated with obesity. The role of such a measure in personalised health and cardiometabolic risk is discussed. Importantly, our work shows a strong association of diet and lifestyle habits with mBMIΔ; higher intake of "healthier foods" such as fruits and fibre and higher levels of leisure time physical activity (PA) were associated with the lower mBMIΔ while prolonged television (TV) viewing time was markedly associated with higher mBMIΔ. This suggests that lifestyle interventions may improve individuals' metabolic health through modification of their mBMI, independent of their measured BMI.

## Results

### Cohort characteristics

AusDiab and BHS are longitudinal, Australian, adult population cohorts. As such, they show similar baseline characteristics, including comparable sex composition, age-, and BMI distribution (Table 1). The prevalence of T2DM, CVD, and smoking were also comparable between the two cohorts. The clinical endpoints in the present study include prevalent (newly diagnosed and untreated) and incident (over a 5-year follow up period) T2DM, pre-diabetes (both prevalent and 5-year incident cases) and incident (over a 10-year follow up period) major cardiovascular events (CVE) and ischemic heart disease (IHD) (Supplementary Tables 1 and 2). The definitions for these outcomes are provided in the method section. The AusDiab and BHS cohorts respectively comprise of 55% and 56% female participants. From the

11,247 AusDiab participants who attended both the interview and the biomedical examinations at baseline, 10,339 had fasting plasma samples available for lipidomic analysis. Of the 10,339 participants, 395 (3.8%) and 291 (2.8%) were identified as newly diagnosed T2DM and known diabetes respectively. Participants with the known diabetes at baseline (i.e., those receiving pharmacological treatment for diabetes, and or previously diagnosed with diabetes) were excluded. During, a 5-year follow up time, 218 incident cases of T2DM were also recorded (Fig. 1a, Supplementary Table 1). In addition, some, 414 major CVEs and 304 IHD (in the AusDiab cohort) (Fig. 1a, Supplementary Table 1) and 284 incident IHD (in the BHS cohort) occurred over 10-year follow up (Fig. 1a, Supplementary Table 2). We examined at the relationship of the anthropometric, clinical and behavioural data in relation to disease outcomes and controls for both cohorts. Most of the explanatory variables were significantly different between cases and controls (Supplementary Table 1 and 2).

### Lipidomic profiling of Australian population cohorts

We utilized previously generated lipidomic data from two large Australian population cohorts, AusDiab[24] and BHS[35]. Targeted lipidomic profiling was performed in each cohort using liquid chromatography coupled to electrospray ionization-tandem mass spectrometry[23], from fasting plasma samples (AusDiab, $n = 10,339$) and fasting serum samples (BHS, $n = 4492$). Lipidomic data encompassing 575 lipid species within 33 lipid classes, from the major glycerophospholipid, sphingolipid, glycerolipid and sterol classes was available on all AusDiab and BHS participants. The coefficient of variation (%CV) of pooled plasma quality control (PQC) samples were calculated for each lipid species to assess the assay performance. In the AusDiab cohort, the median %CV was 10.7% and over 90% of the lipid species were measured with a % $CV < 20\%$[24]. In the BHS cohort, the median %CV was 8.6% with 570 (95.6%) lipid species showing a %CV less than 20%.

### Creation of metabolic BMI scores

We used ridge regression to create a lipidome based predictive model for BMI including age and sex as covariates. To avoid, overfitting, a tenfold cross validation was employed in the AusDiab cohort (i.e., models trained on the 9/10th and used to predict BMI in the holdout 1/10th of the cohort; lambda average = 0.094, range = 0.087–0.105). This model provided predicted BMI (pBMI) values and was able to explain 60.4% of the variance in BMI as shown in Fig. 2a. When the model was validated in the BHS cohort it explained 52.1% of the BMI variance (Supplementary Fig. 1a, Supplementary Table 3). To standardise the pBMI to the population, the metabolic BMI (mBMI) was then derived from the pBMI scores as follows: mBMI = BMI + (pBMI – pBMI value on the line of best fit between pBMI and BMI). The mBMIΔ was then defined as the difference between BMI and mBMI. The correlation between BMI and mBMI was strong: $R^2 = 0.811$ in the AusDiab cohort (Fig. 2b) and $R^2 = 0.71$ in the BHS cohort (Supplementary Fig. 1b). In a sex-specific modelling, metabolic data explained 67% of BMI variation in women and 55% in men (Supplementary Fig. 2). To further assess the precision in estimating mBMI, we generated mBMI scores for the NIST 1950 QC samples (200 replicates, assuming an average BMI of 26.0) that were analysed throughout the AusDiab cohort. The %CV for mBMI in the NIST 1950 QC samples was 5.5%. When we created models using (1) just the clinical lipid measures (total cholesterol, HDL-C and triglycerides) with age and sex, and (2) cardiometabolic risk factors (CMRs, clinical lipids plus HBA1C, FBG, 2h-PLG, HOMA-IR SBP and DBP) the models respectively explained only 15.6% and 31.6% variation in BMI in the AusDiab cohort (Supplementary Table 3, Supplementary Fig. 3) and 10.4% and 31.2 of BMI when validated in the BHS cohort (Supplementary Table 3). We opted to exclude LDL-C from the clinical lipid panel as it's a calculated measure from total cholesterol and triglyceride levels[36].

**Table 1 | Baseline characteristics of the AusDiab and the BHS participants**

| Characteristics | AusDiab ($n = 10,339$) | BHS ($n = 4492$) |
|---|---|---|
| Age (years)[a] | 51.3 (14.3) | 50.8 (17.4) |
| Sex, n (%men)[b] | 4654 (45) | 1976 (44.0) |
| Ethnicity (%White/European ancestry) | 9786 (94.7) | 4492 (100) |
| BMI (kg/m$^2$)[a] | 26.9 (4.9) | 26.2 (4.2) |
| WC (cm)[a] | 90.8 (13.8) | 86.1 (12.7) |
| Cholesterol (mmol/L)[a] | 5.7 (1.1) | 5.6 (1.1) |
| HDL-C (mmol/L)[a] | 1.44 (0.4) | 1.39 (0.39) |
| Triglycerides (mmol/L)[c] | 1.28 (0.9) | 1.18 (0.90) |
| SBP (mmHg)[a] | 129.2 (18.6) | 124.0 (17.9) |
| DBP (mmHg)[a] | 70.0 (11.7) | 74.5 (10.2) |
| FBG (mmol/L)[a] | 5.3 (1.1) | 5.0 (1.4) |
| 2h-PLG (mmol/L)[a] | 6.3 (2.7) | – |
| HbA1C (%)[a] | 5.2 (0.6) | – |
| HOMA-IR[a] | 3.6 (2.4) | 1.78 (2.5) |
| Current smoking, n (%)[b] | 1623 (15.9) | 608 (13.5) |
| BP treatment, n (%)[b] | 1577 (15.3) | – |
| Lipid lowering medication, n (%)[b] | 871 (8.4) | 108 (2.4) |
| Diabetes at baseline, n (%)[b] | 686 (6.6) | 271 (6.0) |
| Baseline CVD prevalence, n (%)[b] | 577 (5.6) | 238 (5.3) |

WC waist circumference, HDL-C high density cholesterol, SBP systolic blood pressure, DBP diastolic blood pressure, FBG fasting blood glucose, 2h-PLG 2-h post load glucose, HbA1C% percent glycated haemoglobin, HOMA-IR homeostasis model assessment of insulin resistance.
[a]Values expressed as mean (±SD).
[b]Values expressed as frequency, n (%) for dichotomous variables.
[c]Data in Median, (IQR) as Triglyceride distribution was right skewed.

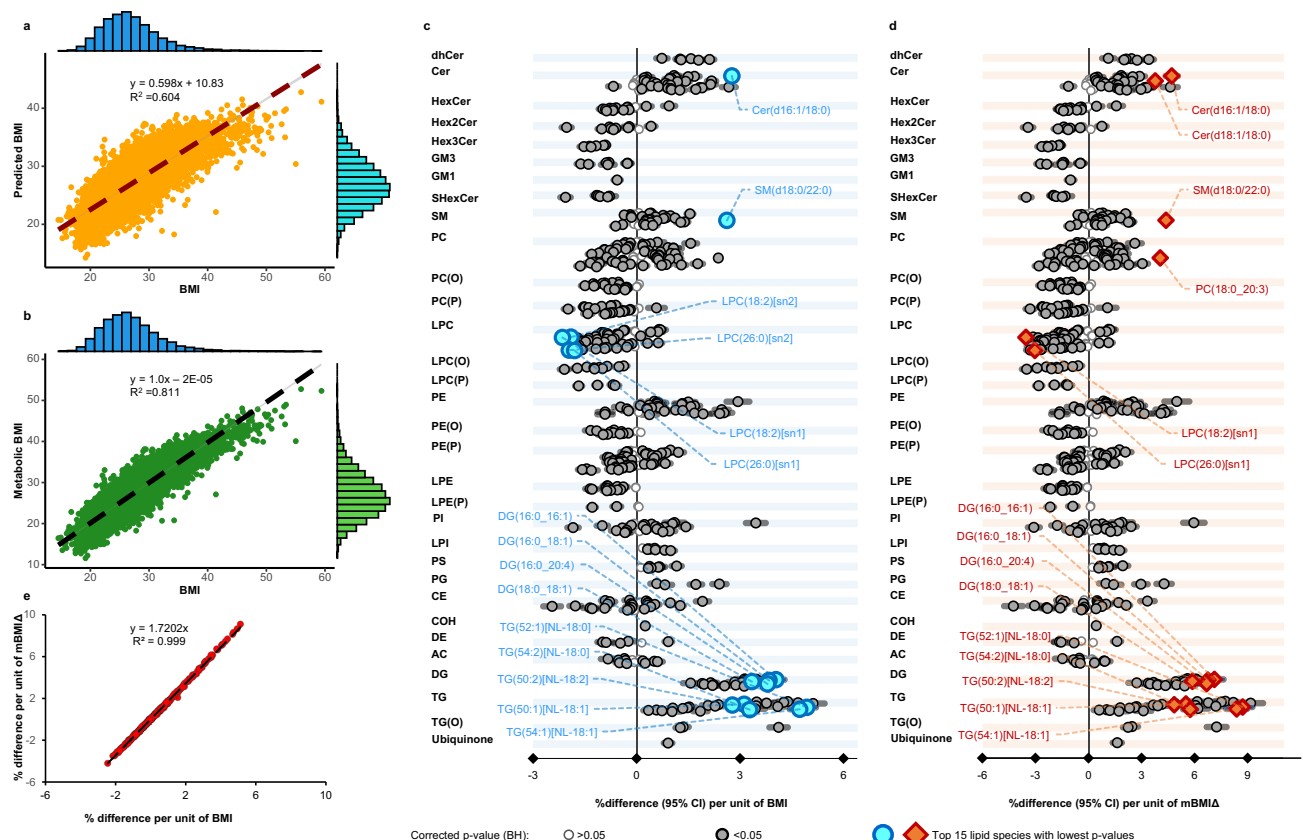

**Fig. 2 | Modelling of the metabolic BMI score and comparison of the captured lipid biology with BMI in the AusDiab cohort (n = 10,339 independent samples). a** Correlation between measured BMI and predicted BMI (orange scatter plot). The blue and sky-blue histogram depicts the distribution of BMI and predicted BMI respectively. **b** Correlation between measured BMI and metabolic BMI (mBMI) (green scatterplot). The blue and green histogram depicts the distribution of BMI and metabolic BMI respectively. **c** Associations of BMI (as a predictor) with plasma lipid species (as outcome, n = 575 species) and (**d**) association of mBMIΔ with plasma lipid species using linear regression analysis adjusting for age and sex. Two-sided *p* values for each lipid species with grey open circles (*p* > 0.05), grey and dark closed circles (*p* < 0.05) are presented after correction for multiple comparisons using the method of Benjamini and Hochberg. Blue circles and brown diamonds represent the top 15 most significant lipid species associated with BMI ($p < 10^{-217}$) and mBMIΔ ($p < 10^{-157}$), respectively. Each data point in (**c**) and (**d**) represent coefficients (% differences) per unit of BMI (**c**) or mBMIΔ (**d**) and the error

bars represent 95% confidence intervals (CI). **e** The correlation between effect sizes of each lipid associated with BMI (x-axis) and with mBMIΔ (y-axis). Additional details are shown in Supplementary Data 1 and 2. AC acylcarnitine, CE cholesteryl ester, Cer ceramide, COH cholesterol, DE dehydrocholesterol, dhCer dihydroceramide, DG diacylglycerol, GM1 GM1 ganglioside, GM3 GM3 ganglioside, HexCer monohexosylceramide, Hex2Cer dihexosylceramide, Hex3Cer trihexosylceramide, LPC lysophosphatidylcholine, LPC(O) lysoalkylphosphatidylcholine, LPC(P) lysoalkenylphosphatidylcholine, LPE lysophosphatidylethanolamine, LPE(P) lysoalkenylphosphatidylethanolamine, LPI lysophosphatidylinositol, PC phosphatidylcholine, PC(O) alkylphosphatidylcholine, PC(P) alkenylphosphatidylcholine, PE phosphatidylethanolamine, PE(O) alkylphosphatidylethanolamine, PE(P) alkenylphosphatidylethanolamine, PG phosphatidylglycerol, PI phosphatidylinositol, PS phosphatidylserine, SHexCer sulfatide, SM sphingomyelin, TG triacylglycerol, TG(O) alkyl-diacylglycerol.

## The biological signal captured by the metabolic BMI

To better understand the lipid biology captured by the mBMI, we performed regression analysis of lipid species with BMI and mBMIΔ. In age and sex adjusted models, we observed a significant association with 505 out of 575 lipid species with BMI. Diacylglycerol, triacylglycerol and ceramide species showed a strong positive association, while most hexosylceramide, lyso and ether phospholipid species were negatively associated (Fig. 2c, Supplementary Data 1) (e.g. LPC(18:2)[sn1] decreased by 2.15% per unit increase in BMI, $p = 1.56 \times 10^{-245}$). Of the triacylglycerol species, TG(52:1)[NL-18:0] was the strongest predictor (4.94% increased per unit of BMI, $p = 4.56 \times 10^{-283}$). We then performed the same regression analysis of lipid species against mBMIΔ (Fig. 2d, Supplementary Data 2) and compared the lipidomic profile associated with BMI with the profile associated with mBMIΔ. Interestingly, the association of mBMIΔ with lipid species and the association of BMI with lipid species were almost identical with the correlation between effect sizes of each lipid associated with BMI (x-axis) and mBMIΔ (y-axis) having a $R^2 = 0.999$. However, we note the effect sizes were stronger against mBMIΔ (Fig. 2e, Supplementary Data 2) reflecting that variance

in mBMIΔ is completely explained by the lipid species whereas variance in BMI is only partially explained by lipid species. For example, the effect size for TG(52:1)[NL-18:0] was 4.94% against BMI (Fig. 2c, Supplementary Data 1) and 8.7% for the same species against mBMIΔ (Fig. 2d, Supplementary Data 2). The statistical explanation why the plot of the beta coefficients of lipids for BMI and mBMIΔ are correlated is elaborated in Supplementary Note 1. A LASSO model performed nearly the same as the ridge model (Supplementary Fig 4a and 4b, Supplementary Table 3). Using the LASSO model, associations of BMI with plasma lipid species (Supplementary Fig. 4c) and association of mBMIΔ with plasma lipid species (Supplementary Fig. 4d) were identical after adjusting for age and sex. The correlation between effect sizes of each lipid associated with BMI (x-axis) and mBMIΔ calculated from the LASSO model (y-axis) provided an $R^2$ close 1.0 (Supplementary Fig. 4e).

## The performance of different regularized linear models to predict BMI

To assess the importance of the number of lipid species in the models, we compared regularized linear models (ridge, elastic-net and LASSO),

incorporating lipid species, age and sex, for their ability to predict BMI in the AusDiab cohort and validated these in the BHS. Using elastic-net (384 lipid species selected) and LASSO (349 lipid species selected) models, we observed similar performance as for the ridge model for the prediction of BMI, with models explaining 60.8% to 60.9% of BMI variance in the AusDiab. Validation of these models in the BHS dataset explained to 52.2% and 51.9% of the BMI variance, compared to 52.1.0% with the ridge model. When we utilised clinical lipids, age and sex in the model development, the elastic-net and LASSO models respectively explained only 15.5% to 15.6% BMI variance in AusDiab and only 10.0% and 10.2% BMI variance in the BHS cohort (Supplementary Table 3). Upon incorporating all CMRs, the elastic net and LASSO models respectively explained, 31. 6% and 31.5 variation in BMI in the AusDiab and 31.1 and 31.2% in the BHS cohort (Supplementary Table 3). As, LASSO and elastic-net showed very similar performance we focused further analysis on the ridge and LASSO models only. To investigate how a further reduction in the number of lipid species in the model affected model performance, we tuned the regularization parameter, lambda, in the LASSO models and in the ridge models for comparison, with log10 lambda values between -4 and 0.2 (Fig. 3a–c). As lambda was increased, the number of features selected into the LASSO model decreased until only 9 lipids are included in the model with a log10 lambda of 0.

In the LASSO models, as lambda increased, the correlation ($R^2$) between BMI and the pBMI decreased, while in the ridge models the $R^2$ remained relatively stable (Fig. 3b). The correlation ($R^2$) between BMI and mBMI increased in the LASSO models reaching a $R^2$ of 1.0 as the number of features incorporated into the LASSO models decreased to 0, but again showed little variation in the ridge models (Fig. 3b). Optimization of the lambda parameter by minimizing the mean-squared error (MSE) using cv.glmnet showed the cross-validated MSE increasing in the LASSO models but again relatively stable in the ridge models (Fig. 3c). The optimum lambda used to model BMI for the ridge and LASSO models was defined by the lowest MSE. We then extracted the beta-coefficients of the optimum ridge and LASSO models (Fig. 3d, e): the lipid species showing the strongest contribution in the ridge and LASSO models were similar. SM(d18:2/14:0), displayed the strongest positive effect size in both models, $\beta = 1.677$ (ridge) and $\beta = 3.172$ (LASSO). More details on the weighting of the individual lipid species in both the ridge and LASSO models can be found in Supplementary Data 3.

While the ridge and LASSO models showed comparable performances, when lambda was optimised, the ridge model was more stable across all the possible lambda values and showed better validation in the BHS cohort (Supplementary Table 3) and so was used for further analyses.

## The association of mBMIΔ with metabolic traits
We hypothesized that the difference between the mBMI the BMI; the mBMIΔ captures cardiometabolic health/risk and this potentially offers clinically relevant information to identify high risk individuals. To assess the relationship between mBMIΔ and cardiometabolic risk factors and explore whether mBMIΔ identifies metabolic subtypes, we grouped the AusDiab participants into quintiles of the mBMIΔ, with just over 2000 participants in each (Fig. 4a). The distributions of BMI and mBMI for the 5 groups are shown in Fig. 4b, c, respectively. We performed linear regression analysis between cardiometabolic traits (outcome) and the quintiles of mBMIΔ (predictor) to assess the overall association. Quintiles 1 to 5 (Q1-Q5), as expected, have comparable BMI values, but substantially different mBMIs. The two most discordant groups (Q1) and (Q5) had similar mean BMI and mean age, while their mBMI scores were significantly different (Fig. 4b–d). The median (IQR) mBMI values were 30.6 (5.5) and 22.7 (6.9) for the Q5 and Q1 respectively. Individuals in Q5 were characterized by unfavourable lipoprotein profiles (higher total cholesterol, higher triglycerides, and lower HDL-C; Fig. 4d), as well as being more insulin resistant, having higher

2-h post-load glucose (2h-PLG), glycated haemoglobin C (HBA1C) and higher blood pressure compared to individuals in Q1 (Fig. 4e), despite Q5 and Q1 having similar mean BMI. We also observed stronger associations of mBMIΔ with waist circumference (WC) and waist-to-hip-ratio (WHR) that with BMI itself (Supplementary Fig. 5), suggesting that these measures are more closely linked to the metabolic dysregulation captured by the mBMIΔ.

To validate these findings, we statistically tested whether the profile of cardiometabolic traits differ between the two most discordant groups in the AusDiab cohort and validated this in the BHS cohort. We performed linear regression analyses (with cardiometabolic traits as outcomes and the discordant groups as the predictor, using Q1 as the reference group), adjusting for age, sex and BMI or for age, sex, BMI, and clinical lipids (excluding the outcome). All the metabolic traits, except FBG, differed between the discordant groups before and after adjusting for clinical lipids despite these groups having a similar BMI in both cohorts (Fig. 5a, b, and Supplementary Table 4). Individuals in Q5 relative to those in Q1 had statistically significantly elevated levels of triglycerides (fold difference 95% CI = 1.52, 1.45–1.59), HOMA-IR (fold difference 95% CI = 1.59, 1.50–1.68) and 2h-PLG (fold difference 95% CI = 1.17, 1.15–1.19). These associations remained significant after further adjustment for clinical lipids, although the effect size was reduced in most cases (Fig. 5b, Supplementary Table 4). The findings observed in the AusDiab cohort were validated on the BHS cohort (note, the 2h-PLG and the HbA1c measures were not available in the BHS cohort). Individuals in the top quintile (Q5) had a significantly elevated level of triglycerides (fold difference 95% CI = 1.44, 1.40–1.49), HOMA-IR (fold difference 95% CI = 1.45, 1.41–1.50) and lower HDL-C (fold difference 95% CI = 0.86, 0.85–0.87) relative to those in the bottom quintile (Q1) (Fig. 5a). These associations remained significant after adjustment with clinical lipids (Fig. 5b).

## Higher metabolic BMI is associated with higher odds of prevalent and future T2DM and pre-diabetes
We assessed the odds of T2DM and pre-diabetes across the quintiles of the mBMIΔ with Q1 as a reference. Individuals with T2DM had higher BMI (mean ± SD = 29.9 ± 6.1) (Fig. 6a) and mBMI (mean ± SD = 31.0 ± 6.0) (Fig. 6b) relative to NGT (mean ± SD BMI = 26.2 ± 4.5 and mBMI = 26.1 ± 5.1). Based on the quintile analyses, there was a progressive increase in the odds ratio of T2DM from the lowest mBMIΔ range (Q1) to the highest (Q5) (Fig. 6c). Individuals in Q5 relative to Q1 had more than four-fold higher odds for prevalent T2DM (OR 95% CI = 4.5, 3.1–6.6, $p$ value = $1.48 \times 10^{-15}$) (Fig. 6c, Supplementary Table 5) and 2.5-fold higher odds for incident T2DM (Fig. 6c, $p = 2.45 \times 10^{-4}$) after adjusting for age, sex, and BMI. These associations were only slightly attenuated but remained significant after adjusting for clinical lipids (total cholesterol level, HDL-C, triglycerides), familial history of diabetes and smoking status. Further details of these associations are provided in Supplementary Table 8. We have previously reported comprehensive sex-differences in the lipidomic profile employing the same datasets[24]. Recognizing these differences in the metabolic profiles of men and women, we conducted a separate analysis for men and women. In sex-stratified models (age and BMI adjusted), we observed that, the mBMIΔ exhibited a slightly stronger association with a newly diagnosed T2DM in women than men (Supplementary Fig. 6).

Next, we investigated whether the strong associations of mBMIΔ with T2DM observed above also exist in the pre-diabetic state. We performed a logistic regression between mBMIΔ quintiles and prevalent pre-diabetes ($n = 1920$) versus NGT ($n = 7733$) or 5-year incident pre-diabetes ($n = 417$) versus NGT controls ($n = 4023$, those who remained NGT over the follow up period). As the mBMIΔ increased, the odds of pre-diabetes at baseline and risk of future pre-diabetes increased in a progressive manner. Subjects in the top quintile of mBMIΔ (Q5), despite having a BMI comparable to those in the Q1, had a threefold higher odds of prevalent pre-diabetes (OR 95% CI = 3.0,

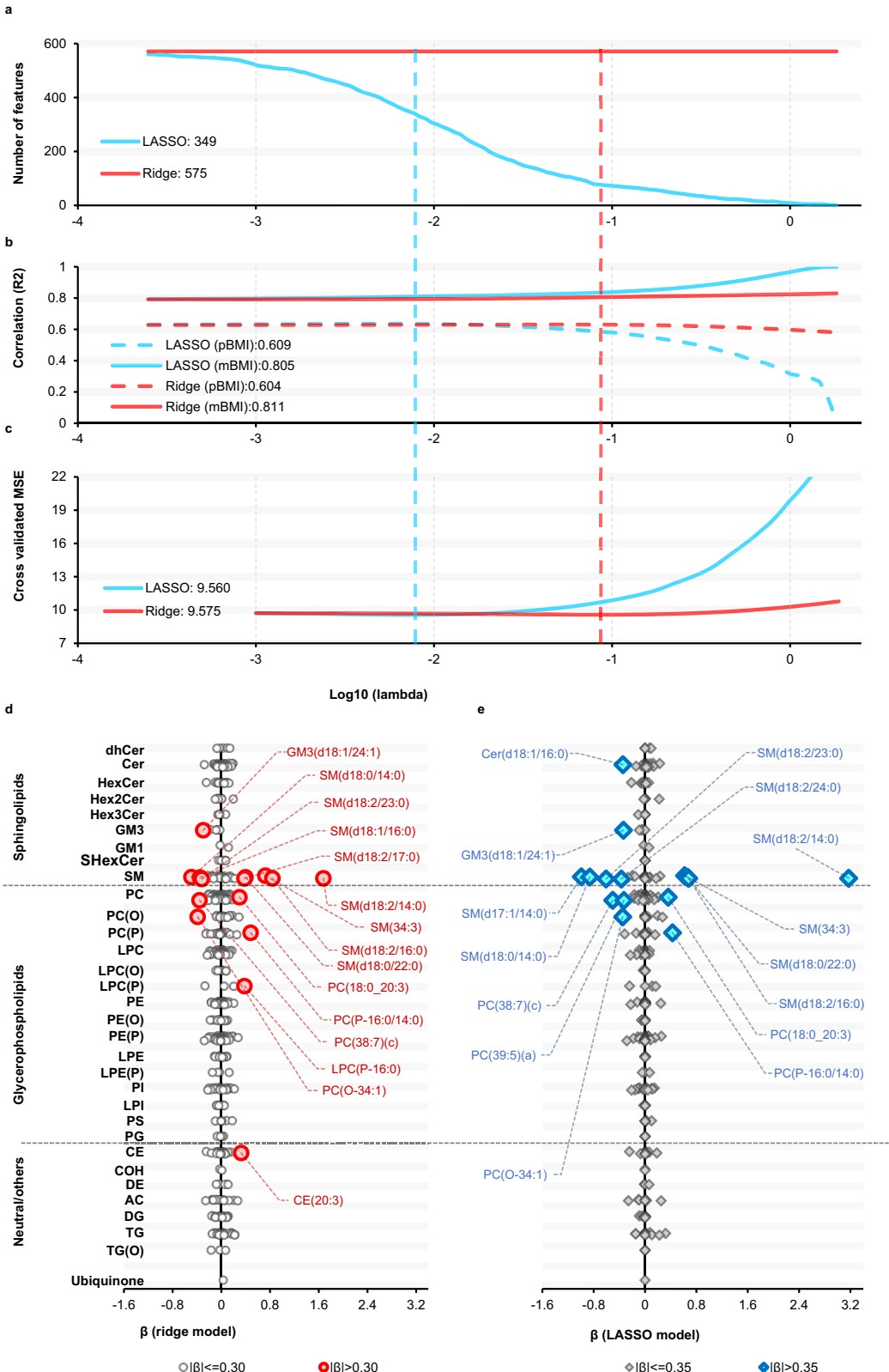

2.5–3.5, $p = 1.54 \times 10^{-33}$) compared to those belonging to the lowest quintile of mBMIΔ. In addition, subjects in Q5 with NGT at baseline had more than a twofold higher odds of progressing to pre-diabetes prospectively compared to those in the Q1 (OR 95% CI = 2.5, 1.8–3.5, $p$ value = $3.67 \times 10^{-8}$). This association remained significant (although attenuated) upon adjusting for clinical lipids, and smoking (Fig. 7, Supplementary Table 6). The details of the odds ratios and p-values

before and after adjusting for clinical lipids across the full quintile range are provided in Supplementary Table 6. Prevalent pre-diabetes constitutes two distinct pre-diabetic states: isolated impaired fasting glucose (IFG) and impaired glucose tolerant (IGT) and the composite of these two. The association of mBMIΔ with isolated IGT was stronger than the association with IFG. However, in both cases a strong and progressive increase in the odds ratio was observed as one moves from

**Fig. 3 | The performance of ridge and LASSO models. a** The number of features incorporated in the ridge (red line) and LASSO (blue line) models for different lambda values. **b** The correlation ($R^2$) of BMI and pBMI (dashed lines) or BMI and mBMI (solid lines) in ridge (red line) and LASSO models (blue line) for different lambda values. **c** MSE of the difference between the observed and predicted values for ridge (red line) and LASSO models (blue line). The vertical dashed red and blue lines represent the minimum MSE, for ridge and LASSO models respectively (i.e., the optimum lambda used to make the models). **d** A plot of beta coefficients from the optimum ridge model. **e** A plot of beta coefficients from the optimum LASSO model. Red circles and blue diamonds represent the top 15 lipid species (ranked based on the absolute value of beta coefficients) showing the strongest contribution in the ridge and LASSO models respectively. AC acylcarnitine, CE cholesteryl

ester, Cer ceramide, COH cholesterol, DE dehydrocholesterol, dhCer dihydrocer-amide, DG diacylglycerol, GM1 GM1 ganglioside, GM3 GM3 ganglioside, HexCer monohexosylceramide, Hex2Cer dihexosylceramide, Hex3Cer trihexosylceramide, LPC lysophosphatidylcholine, LPC(O) lysoalkylphosphatidylcholine, LPC(P) lysoalkenylphosphatidylcholine, LPE lysophosphatidylethanolamine, LPE(P) lysoalkenylphosphatidylethanolamine, LPI lysophosphatidylinositol, PC phospha-tidylcholine, PC(O) alkylphosphatidylcholine, PC(P) alkenylphosphatidylcholine, PE phosphatidylethanolamine, PE(O) alkylphosphatidylethanolamine, PE(P) alke-nylphosphatidylethanolamine, PG phosphatidylglycerol, PI phosphatidylinositol, PS phosphatidylserine, SHexCer sulfatide, SM sphingomyelin, TG triacylglycerol, TG(O) alkyl-diacylglycerol. Source data are provided as a Source Data file.

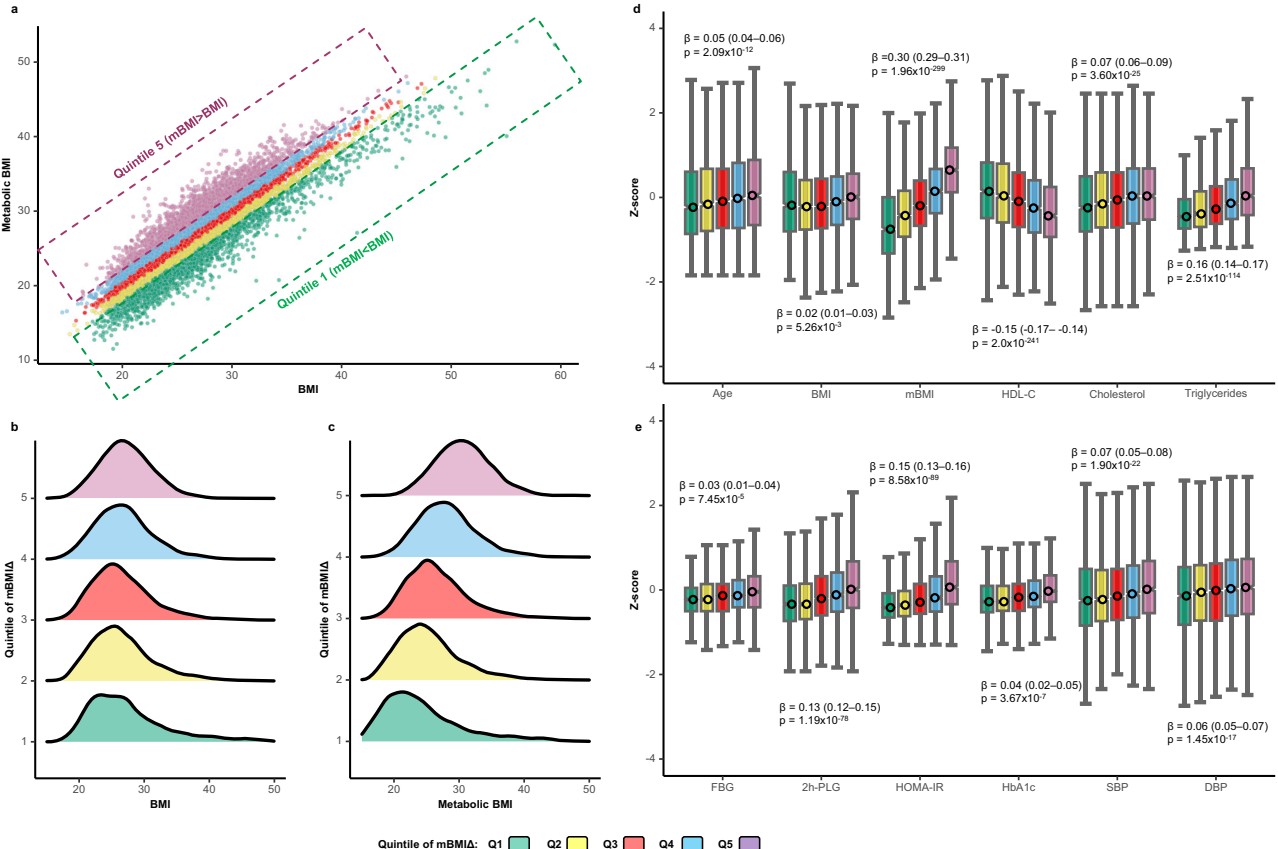

**Fig. 4 | The relationship between mBMIΔ and cardiometabolic traits.**
**a** Correlation between mBMI and BMI for all individuals across the quintiles of mBMIΔ in the AusDiab dataset ($n = 10,339$). The green, yellow, red, blue, and pink marks show individuals in the Q1 ($n = 2068$), Q2 ($n = 2068$), Q3 ($n = 2067$), Q4 ($n = 2068$) and Q5 ($n = 2068$) of mBMIΔ respectively. **b** Density histograms of BMI distribution for each mBMIΔ quintile. **c** Density histograms of mBMI distribution for each mBMIΔ quintile. **d, e** Box plots of the association of mBMIΔ with cardio-metabolic traits. Box plots represent the distribution of z-scores of the respective cardiometabolic trait in each quintile of mBMIΔ. The data depicted in the box and whisker plots for (**d**) and (**e**) span from the minimum to the maximum values

(z-score). The lower and upper boundaries of the box correspond to the 25th and 75th percentiles, respectively, and the central open circles within the boxes represent the median values. Linear regression analyses of mBMIΔ quintile (pre-dictor) against cardiometabolic traits (outcome) were performed. β-coefficients and $p$ values (two-sided) from the linear regression analyses are presented. No adjustments were made for multiple comparisons. BMI body mass index, HDL-C high density cholesterol, HOMA-IR homeostatic model assessment of insulin resistance, FBG fasting blood glucose, 2h-PLG 2-h post load glucose, SBP systolic blood pressure, DBP diastolic blood pressure, HbA1C haemoglobin A1c. Source data are provided as a Source Data file.

Q1 to Q5 of mBMIΔ (Supplementary Fig. 7). A significant association exists between the mBMIΔ and the isolated IFG versus NGT, despite the weak association of mBMIΔ with FBG itself. We identified that, the latter finding (i.e., weak associations of mBMIΔ with FBG) resulted from the presence of subjects with very high FBG levels and known diabetes mellitus (KDM) in the whole cohort (Supplementary Fig. 8). Of note, individuals with KDM has a lower mBMIΔ than those with IFG, IGT and NGT (Supplementary Fig. 8). The associations of mBMIΔ with IFG were independent of 2h-PLG and associations with IGT were inde-pendent of FBG (Supplementary Table 7).

## Higher metabolic BMI tracks the risk of CVD

We assessed whether the mBMIΔ was associated with prevalent CVD and risk of future CVE independent of the measured BMI. Individuals in the top mBMIΔ quintile, Q5 were twice as likely to have prevalent CVD relative to those in the lowest quintile, Q1 (OR 95% CI = 2.1, 1.5–3.1, $p = 6.43 \times 10^{-5}$) (Table 2). Additional adjustment for total cholesterol, HDL-C, triglycerides, smoking status, and family history of diabetes did not attenuate mBMIΔ/mBMIΔ quintile–prevalent CVD associations (Supplementary Table 8). Compared to the entire cohort, the results were consistent in a sex-stratified analyses, although the mBMIΔ

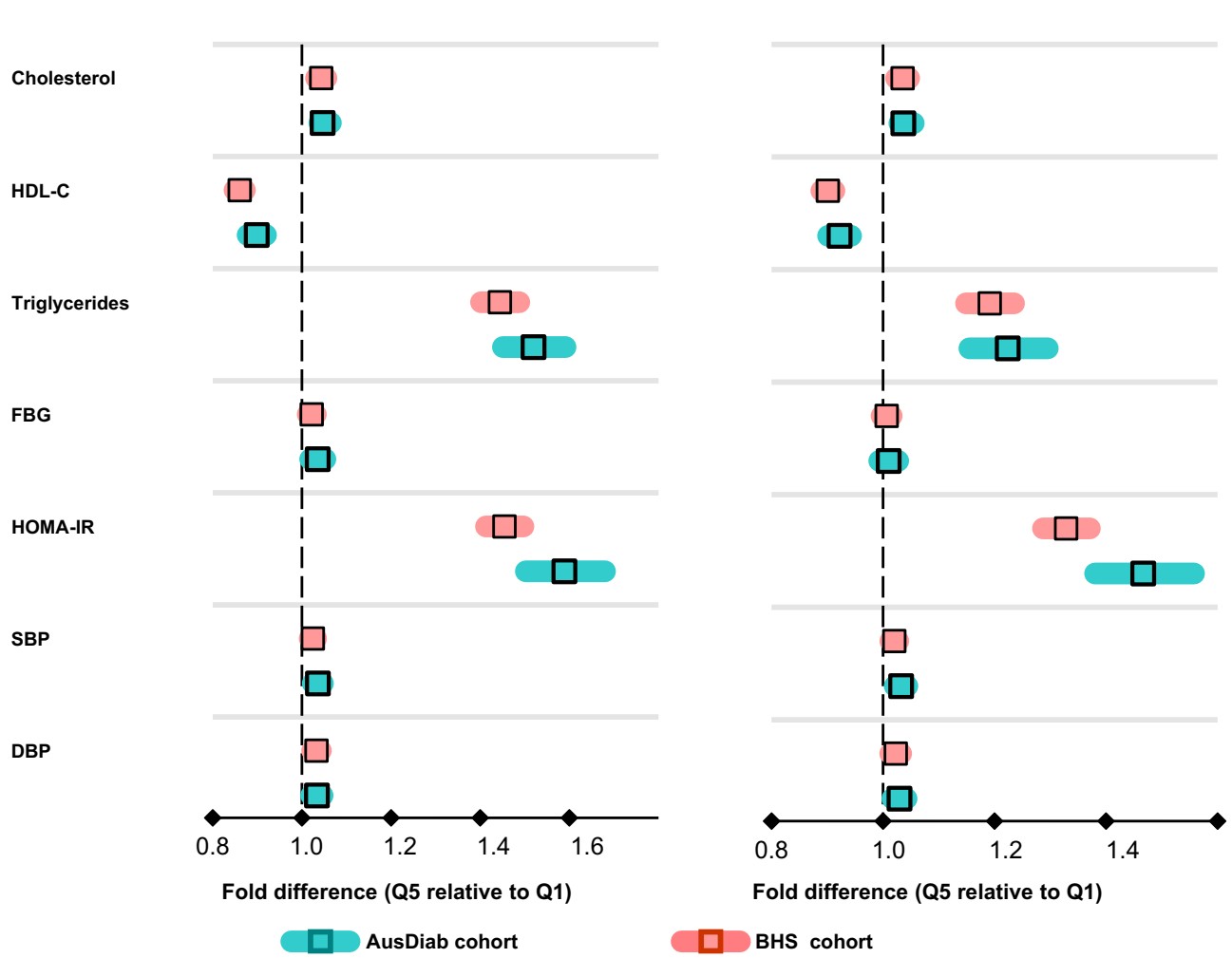

**Fig. 5 | Validation of the association of cardiometabolic risk factors with metabolic discordant groups.** Linear regression analyses between metabolic traits (outcomes) and the discordant mBMIΔ groups (predictor, Q5 relative to Q1) were performed adjusting for (**a**) age, sex, and BMI and (**b**) age, sex, BMI, total cholesterol, HDL-C, and triglycerides (excluding the outcome) in the AusDiab cohort, n = 10, 339 (blue green boxes) and the BHS cohort, n = 4492 (pink boxes). Each square represents the fold difference (Q5 relative to Q1 of mBMIΔ) for a given metabolic trait. The whiskers represent 95% CIs. HDL-C high density cholesterol, HOMA-IR homeostatic model assessment of insulin resistance, FBG fasting blood glucose, SBP systolic blood pressure, DBP diastolic blood pressure.

exhibited a slightly larger effect size in women (Supplementary Fig. 9, middle panel) than men (Supplementary Fig. 9, bottom panel). The mBMIΔ was only marginally associated with the 10-year major incident CVE (HR 95% CI = 1.11, 1.01–1.22, $p = 4.3 \times 10^{-2}$) (Table 2) and IHD event (HR 95% CI = 1.13, 1.01–1.27, $p = 3.6 \times 10^{-2}$) (Table 2). Only the, IHD events in the AusDiab were defined in the same way in the BHS. Consequently, we validated the mBMIΔ–IHD associations in the BHS cohort showing similar results as in the AusDiab (Supplementary Table 9).

In a sensitivity analyses, a mBMIΔ calculated from the model using clinical lipids exhibited a strong correlation (r = 0.68) with the mBMIΔ calculated using the model incorporating CMRs. However, the correlation between mBMIΔ derived from the clinical lipid values or the CMRs and the score calculated using lipidomic data (the current mBMIΔ) were weaker, (R = 0.3 and 0.34 respectively) (Supplementary Fig. 10a). This suggests that the lipidomic data capture independent information that is not fully accounted for by the clinical lipid values or the CMRs alone. We also examined how the mBMIΔ derived from clinical lipids or CMRs related with disease outcomes and how these compared with the current mBMIΔ. Although, the BMI prediction performance was low when using only clinical lipids or CMRs

(Supplementary Fig. 3), the mBMIΔ calculated from clinical lipid values performed as well as the mBMIΔ of the lipidome model in predicting the prevalent T2DM (Supplementary Fig. 10b) and incident T2DM (Supplementary Fig. 10c). As expected, mBMIΔ calculated from the CMRs model performed better than the lipidomic model at prediction of prevalent and incident T2DM as the diagnostic criteria are included in the model. In contrast the mBMIΔ derived from model with clinical lipids or CMRs did not predict cardiovascular disease, demonstrating the limitations of these models (Supplementary Fig. 10d).

**Comparison of models with and without mBMIΔ**

Using mBMIΔ as a continuous outcome, we assessed the relative contribution of BMI and mBMIΔ in models containing both BMI and mBMIΔ adjusting for age and sex in the AusDiab cohort. We also assessed the association of mBMI against the same outcomes. As expected, BMI was strongly associated with both prevalent and incident T2DM and to the lesser extent with prevalent CVD and incident CVE (Supplementary Table 10). The mBMI itself was also significantly associated with T2DM and prevalent CVD independent of age and sex; these associations were stronger (resulting in lower *p* values) than the associations with either measured BMI or mBMIΔ (Supplementary

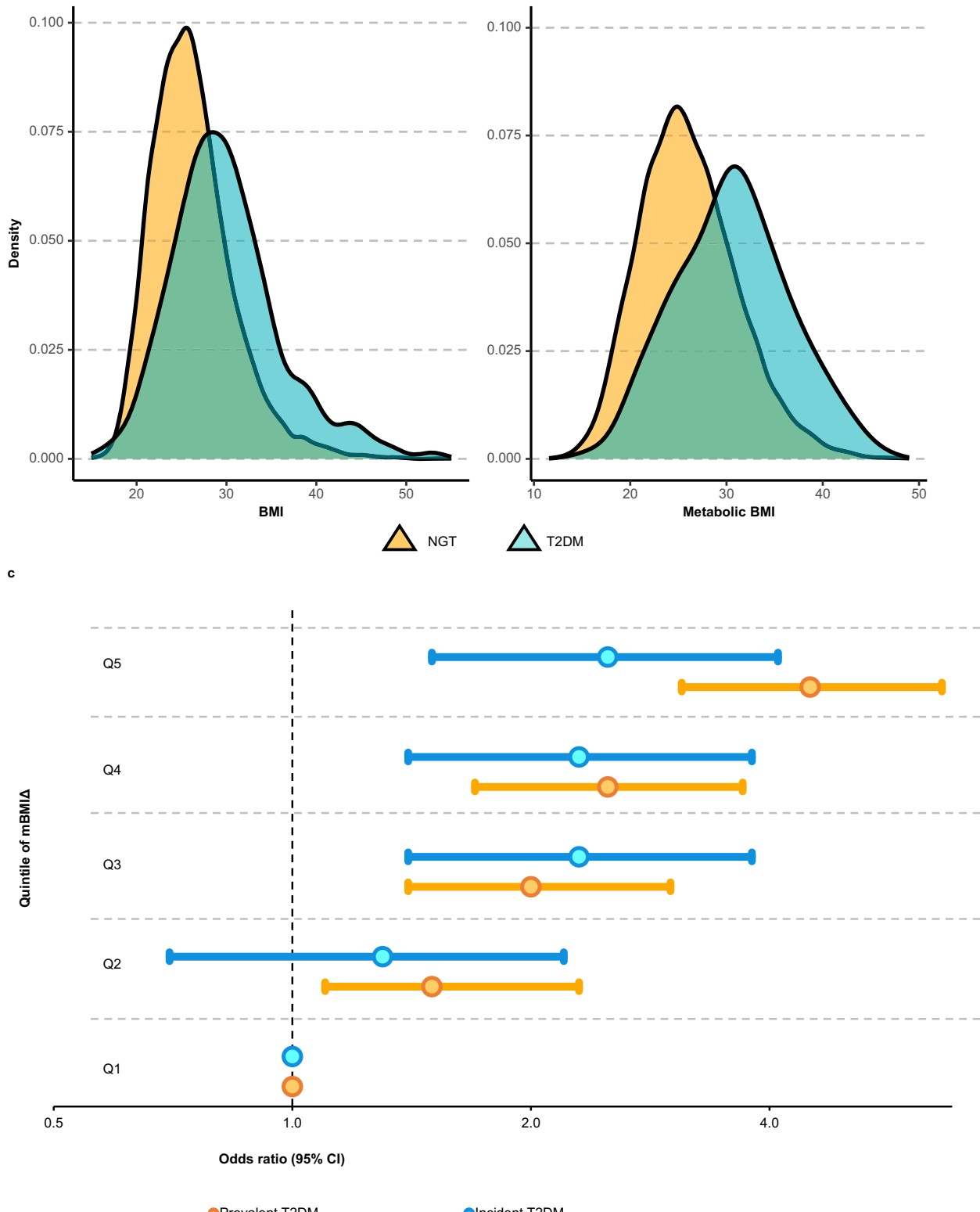

**Fig. 6 | The relationship between mBMIΔ and T2DM. a** Density histogram showing the distribution of BMI in T2DM and NGT subjects. **b** Density histogram showing the distribution of mBMI in T2DM and NGT subjects. **c** The forest plot displays the odds ratio (x-axis) associated with moving from Q1 of mBMIΔ (reference quintile) to Q2–Q5 (y-axis) for the newly diagnosed prevalent T2DM (yellow circles) and 5-year incident T2DM (sky-blue circles) compared to controls. The odds ratios were computed from a multiple logistic regression between a newly diagnosed prevalent T2DM, $n = 395$ versus 7733 NGT subjects at baseline or incident T2DM, $n = 218$ cases versus 5354 controls free of T2DM and the quintiles of the mBMIΔ (Q1 as a reference) adjusted for age, sex, and BMI. Error bars represent 95% CIs. Odds ratios and the associated CIs were log2 transformed to enhance visualization. The results for clinical lipid, familial history of diabetes and smoking status adjusted models are provided in Supplementary Table 5.

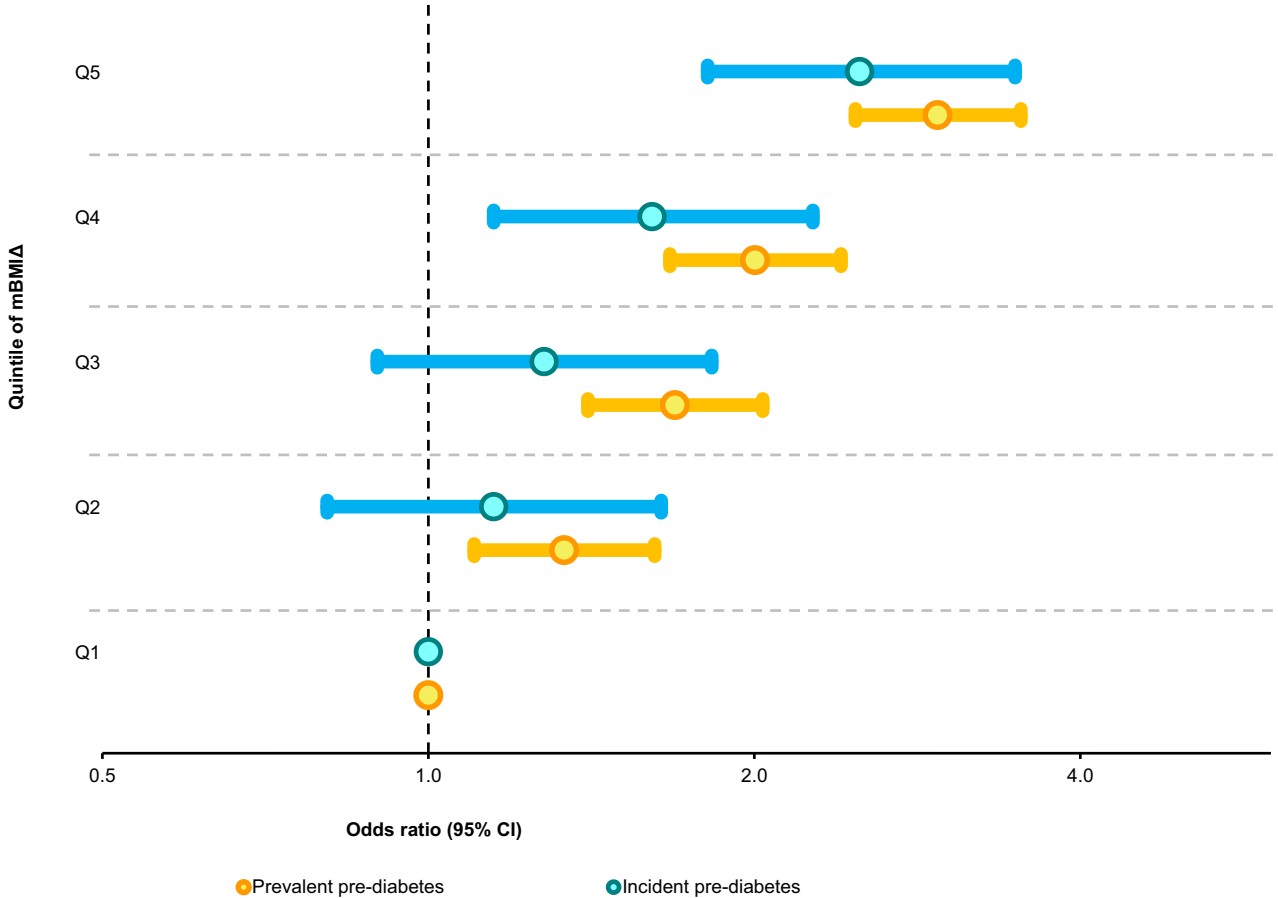

Prevalent pre-diabetes   Incident pre-diabetes

**Fig. 7 | The relationship between mBMIΔ and pre-diabetes.** Depicted on the x-axis of the forest plot are the odds ratios (on a log2 scale) for subjects with the prevalent pre-diabetes (gold circles) and 5-year incident pre-diabetes (sky-blue circles) compared to the controls across the quintiles of mBMIΔ (y-axis). The odds ratios were computed using a logistic regression between prevalent pre-diabetes, $n = 1920/7733$ NGT or incident pre-diabetes, $n = 417/4023$ NGT and the quintiles of the mBMIΔ (Q1 as a reference) adjusted for age, sex, and BMI in the AusDiab cohort. Each circle and the horizontal errors bars (95% CI) for the quintiles (Q2–Q4) represent the odds of pre-diabetes associated with moving from the reference quintile (Q1, OR = 1). Detailed associations including clinical lipids and smoking adjusted analyses are presented in Supplementary Table 6.

**Table 2 | The association between mBMIΔ/quintiles of mBMIΔ and CVD outcomes (prevalent and incident) in the AusDiab cohort**

| mBMIΔ | Prevalent CVD ($n = 577$ cases versus 9690 controls) | | 10-year incident CVE ($n = 414$ events versus 7936 non-events) | | 10-year incident IHD ($n = 304$ events versus 8046 non-events) | |
|---|---|---|---|---|---|---|
| | Odds ratio (95% CI)[a] | *p* value | Hazard ratio (95% CI)[b] | *p* value | Hazard ratio (95% CI)[c] | *p* value |
| mBMIΔ (continuous scale) | 1.3 (1.1, 1.4) | **$3.40 \times 10^{-5}$** | 1.11 (1.01, 1.22) | **$4.30 \times 10^{-2}$** | 1.13 (1.01, 1.2) | **$3.6 \times 10^{-2}$** |
| Q1 (Ref) | – | – | – | – | – | – |
| Q2 | 1.4 (1.01, 2.1) | $7.30 \times 10^{-2}$ | 1.1 (0.8, 1.5) | $7.00 \times 10^{-1}$ | 1.0 (0.7, 1.5) | $9.00 \times 10^{-1}$ |
| Q3 | 1.3 (0.9, 2.0) | $1.70 \times 10^{-1}$ | 1.0 (0.7, 1.3) | $8.00 \times 10^{-1}$ | 0.9 (0.6, 1.4) | $7.00 \times 10^{-1}$ |
| Q4 | 1.7 (1.1, 2.4) | **$1.10 \times 10^{-2}$** | 1.4 (1.02, 1.9) | **$4.00 \times 10^{-2}$** | 1.4 (1.02, 2.1) | **$4.10 \times 10^{-2}$** |
| Q5 | 2.1 (1.5, 3.1) | **$6.40 \times 10^{-5}$** | 1.3 (0.9, 1.7) | $1.30 \times 10^{-1}$ | 1.3 (0.9, 1.8) | $2.00 \times 10^{-1}$ |

Significant two-sided *p* values (<0.05) are shown in bold.
*CVD* cardiovascular disease, *CVE* cardiovascular event, *IHD* ischemic heart disease.
[a]Logistic regression between the mBMIΔ /quintiles of mBMIΔ and prevalent CVD adjusting for age, sex, BMI, smoking status and diabetes history. Odds ratios and 95% CIs for subjects with prevalent CVD compared to control groups were computed.
[b]Proportional hazard Cox-regression between the mBMIΔ /quintiles of mBMIΔ and major incident CVE adjusting for age, sex, BMI, smoking status and diabetes history. Hazard ratios and 95% CIs for subjects with incident CVE compared to non-events are presented.
[c]Proportional hazard Cox-regression between the mBMIΔ /quintiles of mBMIΔ and incident IHD adjusting for age, sex, BMI, smoking status and familial diabetes history. Hazard ratios and 95% CIs for subjects with incident IHD compared to non-events are presented.

Table 9). The mBMIΔ showed, an independent association with prevalent and incident T2D after correcting for age, sex and BMI (Supplementary Table 10) and with CVD outcomes after adjusting for age, sex, BMI, smoking status and diabetes. To assess the significance of the additional information provided by the mBMIΔ to the prediction of T2DM, Akaike's information criterion (AIC) and Likelihood ratio test (LRT) were calculated to compare the two competing nested models (i.e., one containing mBMIΔ the other without mBMIΔ). Using this

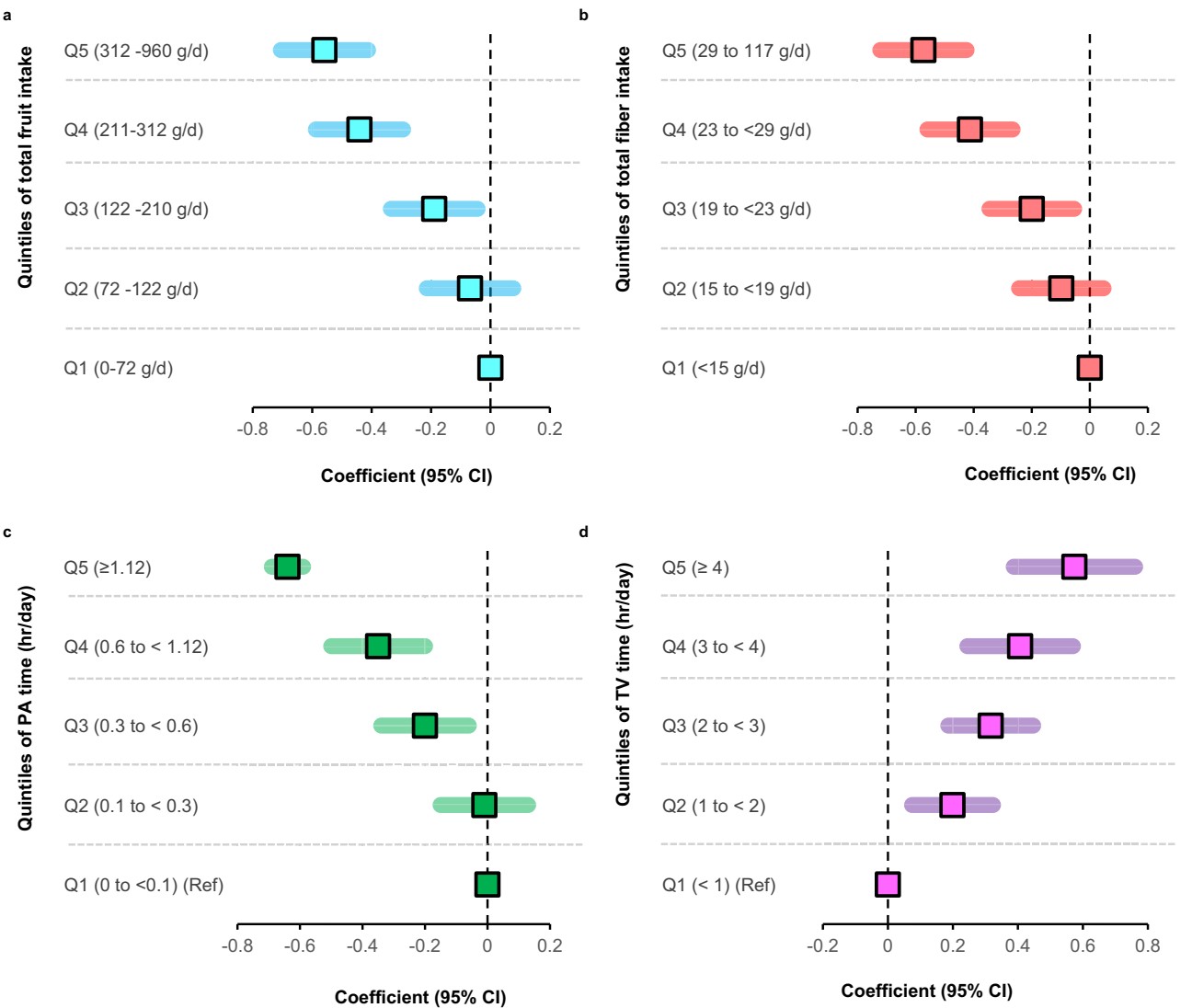

**Fig. 8 | Associations of dietary and lifestyle habits with mBMIΔ.** Forest plots show age, sex and BMI adjusted coefficients (95% CIs) (x-axis) in a multiple linear regression analysis of mBMIΔ against (**a**) the quintiles of total fruit intake (**b**) quintiles of fibre intake (**c**) PA level in hrs/day and (**d**) TV viewing time in hrs/day. Square boxes represent the coefficients (units of mBMIΔ in Kg/m²) associated with moving from the reference quintile (Q1) to Q2 – Q5 of each diet or lifestyle. PA, physical activity; TV, television. Source data are provided as a Source Data file.

approach, we showed that models with mBMIΔ showed a better fit in predicting newly diagnosed prevalent T2DM (i.e., models with mBMIΔ have smaller AIC (AIC = 2603.1) compared to models without mBMIΔ (AIC = 2652.4)) and a LRT $p$ value of $8.02 \times 10^{-13}$. In predicting incident T2DM, the model with mBMIΔ fit significantly better (AIC = 1733.1) than the model without mBMIΔ (AIC = 1742.4) and a LRT $p$ value = $7.98 \times 10^{-4}$. The model with mBMIΔ also showed a better fit for prevalent CVD relative to a model without mBMIΔ (Supplementary Table 11).

**Lifestyle and dietary habits are associated with mBMIΔ**

Using dietary data in the AusDiab (n = 10, 339), we assessed whether certain dietary habits were associated with mBMIΔ. Total fruit intake (quintiles) encompassing 10 different types (Supplementary Fig. 11) and total fibre intake (quintiles) were inversely associated with mBMIΔ. In a model adjusted for age, sex and BMI (model 1), total fruit intake was inversely associated with mBMIΔ (Q5 vs Q1, β −0.56 [95% CI −0.71 to −0.41], $p = 8.54 \times 10^{-14}$) (Fig. 8a). In the full model, adjusted for smoking, PA time, TV viewing time, SBP, family history of diabetes, history of CVD and other dietary and lifestyle factors (model 2), this association remained significant (β −0.25, [95% CI −0.44 to −0.06],

$p = 3.90 \times 10^{-03}$) (Supplementary Table 12). Compared to participants with the lowest intake of total dietary fibre (Q1), participants with the highest intake (Q5) had a 0.57 lower mBMIΔ (β, −0.57; 95% CI, −0.72 to −0.43, $p = 4.36 \times 10^{-14}$) (Fig. 8b). In the full model, this association was only slightly attenuated but remained significant (Supplementary Table 12). A strong dose-response relationship between the quintiles of PA time and mBMIΔ was observed. Participants in Q5 (average PA time, 2 hrs/day) had a 0.64 (β −0.64 [95% CI −0.79 to −0.50], $p = 6.31 \times 10^{-18}$) lower mBMIΔ relative to those in Q1 (average PA time = 0 hrs/day) (Fig. 8c). In the fully adjusted model PA remained significantly associated with mBMIΔ ($P < 0.05$) (Supplementary Table 12). Prolonged TV viewing time was also significantly associated with mBMIΔ. Compared to the Q1 reference category (TV viewing time <1 hr/day), participants in Q5 who spent ≥4 hours/day had a 0.57 higher mBMIΔ (β, 0.57); 95% CI, [0.39–0.76], $p = 1.76 \times 10^{-09}$ (Fig. 8d) and remained significant in the fully adjusted model (Supplementary Table 12).

## Discussion

Obesity is a major risk factor for many non-communicable diseases such as T2DM and CVD[11–13,37]. However, the widely used measure of obesity, BMI, does not fully capture the metabolic dysregulation

associated with obesity leading to the misclassification of metabolic health and metabolic risk. While direct measures of body fat distribution, such as computed tomography and dual-energy X-ray absorptiometry, have the potential to enhance risk assessment by providing valuable insights into body fat distribution, their practical application is constrained by high costs and inability to directly evaluate metabolic health and perturbations. In contrast, mBMI measures hold promise for understanding metabolic health and risk, albeit requiring development of their clinical utility and cost-effectiveness[38]. Hence, in the present study, we constructed a lipidome-based BMI score, that represents the mBMI of an individual, with a view to understand its biological significance and examine whether the score provides additional information over the measured BMI for the metabolic health and risk assessment of multiple clinical outcomes. The mBMI score, although not intended to replace existing risk scores for cardiometabolic diseases, serves as a measure of cardiometabolic health (e.g., to identify individuals who may be metabolically unhealthy despite their BMI). By focusing on metabolic health assessment, the mBMI score provides valuable insights into an individual's metabolic risk profile, enabling targeted interventions through diet and lifestyle modifications to address specific metabolic health concerns prior to the onset of disease. Here, we introduced quintiles of mBMIΔ and stratified the population based on the disparity between BMI and mBMI. We report key associations of mBMIΔ and metabolic discordant groups with cardiometabolic traits, pre-diabetes, T2DM, and CVD after accounting for BMI and other appropriate covariates. In addition, we assessed the relationship of dietary and lifestyle habits with mBMIΔ. We observed that, higher intakes of fruits and fibre or higher levels of PA time were inversely associated with mBMIΔ, while prolonged TV viewing time was associated with higher mBMIΔ.

Lipidomic and metabolomic studies show that BMI is strongly associated with dysregulation in lipid metabolism[21–25,28,39]. To better understand the biology captured by mBMI, we, examined the relationship of the mBMIΔ with the lipidomic profile and compared this with the relationship of BMI with the same lipid species. As previously reported by us and others, most plasma lipid class/subclasses/species were significantly associated with BMI. Glycosphingolipids and phospholipids exhibited predominantly negative associations, while most ceramide, sphingomyelin, diacylglycerol, and triacylglycerol species demonstrated positive associations. The associations of the same lipid species with mBMIΔ were almost identical to the associations with BMI, with the correlation of the coefficients showing a $R^2$ of 0.999. However, the effect size was 1.72-fold greater for the mBMIΔ relative to the associations with BMI. This similarity between the associations of lipid species with BMI and mBMIΔ demonstrates that the mBMIΔ captures the same biology (i.e., dysregulation of lipid metabolism associated with BMI), but captures that portion that is missed (orthogonal to the measured BMI) in the BMI measure. Given the method used to calculate the mBMIΔ, it is not surprising that the correlation between coefficients is close to 1.0. A theoretical description of this relationship is given in Supplementary Note 1. This has important implication as to how we understand and interpret the mBMIΔ and the mBMI itself. It appears that mBMI then, represents the metabolic status of each individual and that this incorporates both the metabolic dysregulation captured by their measured BMI but also the metabolic dysregulation (of the same lipid metabolic pathways) that is not captured by their BMI. It is not surprising then that mBMI provides an improved risk marker compared to BMI itself.

In the present study, our ridge and LASSO models, included 575 lipid species spanning the sphingolipid, phospholipid, glycolipid, and sterol classes along with age and sex as input variables, explained 60.4% and 60.9% of BMI variability respectively (Supplementary Table 3), implying that dysregulation in lipid metabolism is a major consequence of obesity. We included all the measured lipids in the model to determine how well the entire lipidome explains BMI, rather than focusing on only those that were significantly associated with BMI. In previous studies, ridge regression has been used to create mBMI scores using different sets of metabolites[21,33]. A study that used untargeted metabolomic datasets encompassing 650 blood metabolites (47% lipids) and 49 BMI associated metabolites out of the 650 (40% lipids) demonstrated that 49% and 43% of BMI variation was explained by these sets respectively[21]. Using three independent clinical cohorts, a ridge model with 108 plasma metabolites explained BMI variation ranging from 19 to 47%[33]. While with, a LASSO model, a set of 250 randomly selected lipid species were used to model BMI, and these explained 47% of the variation in BMI[22]. In a recent multi-omics study, the application of the LASSO algorithm resulted in the retention of 62 out of 766 metabolites which collectively explained up to 68.9% of the variance in BMI, and the combined model (metabolomics, proteomics and clinical measures) exhibited the further improvement ($R^2$ = 78%)[34]. The difference in the BMI variance explained in these different studies could be related to the range of molecular markers used when modelling BMI, population setting, experimental design and modelling approaches. Generally, models based on limited set of metabolites result in a smaller proportion of the variance in BMI being explained compared to models based on more complex metabolite profiles[21]. Moreover, inherent differences exist in the mBMI scores, stemming from the nature of metabolites used to model BMI across different studies. While our current score is based on lipidomic profiles, other studies[21,33,34] have utilized metabolites from amino acids, carbohydrates, xenobiotics, and lipid metabolic pathways. As a result, the biological information captured by the scores is distinct, although there will be a significant overlap in the detected signal. Indeed, although our LASSO model (containing 349 lipid species) performed equal to the ridge model (containing 575 lipid species), when we further decreased the number of lipid species in the LASSO models by increasing lambda, we observed a decrease in the correlation of pBMI and BMI scores (proportion of variance explained).

Examination of Fig. 3 shows that this effect occurs as the number of lipid species in the model drops below 200 with the correlation decreasing more dramatically as the number decreases below 100. This was associated with an increase in the mean square error (MSE) of the models. Increasing lambda did not have the same effect in the ridge models where all lipid species were retained in the models. These results suggest a minimum number of lipid species (100-200) are required to capture the maximum variance in BMI and so provide an optimal mBMI score. We recognise that the number of lipid species will also be dependent on the species themselves, their association with BMI and the quality of the measurements. In this later regard, models based on targeted lipidomic profiling as used here may offer some advantages over models based on untargeted metabolomics[21] and shotgun lipidomics[22]. Notwithstanding these dependencies, we observe that the coefficients in the optimal ridge and LASSO models were very similar with many of the strongest lipids identical between models and the weighting structure showing similarities across lipid classes (Figs. 3d and 3e). We note the prominent role of sphingomyelin species, such as SM(d18:2/14:0), in the models as illustrated in the Figs. 3d and 3e. Furthermore, it is worth noting that these figures highlight he lipid species that make the greatest contribution to the ridge and LASSO models with species of sphingomyelin and several phospholipid classes playing a prominent role.

Despite its simplicity and convenience, BMI alone does not capture the myriad of obesity related health consequences[40]. Prior evidence suggests that people with the same or similar BMI can display a substantial difference in their metabolic health outcomes[41,42]. A specific group of individuals who fall within the normal BMI range but exhibit indicators of cardiovascular risk, including insulin resistance, elevated triglyceride levels, and coronary heart disease has been identified[43,44]. There are also overweight or obese individuals, based on their BMI, who are metabolically healthy (MHO)[45,46], although the vast

majority of these convert to metabolically unhealthy obese over time[47]. Indeed, it is also crucial to acknowledge that BMI does not account for ethnic differences, lifestyle factors, and muscle mass. Consequently, certain populations such as Asians face a heightened risk of cardio-metabolic disease compared to white Europeans at the same BMI[48]. In these cases, a relatively higher BMI/WC cut-off point might be warranted to accurately screen for diabetes and metabolic syndrome[49]. Similarly, in case of professional athletes, high BMI overestimates adiposity due to the increased muscle mass. Thus, relying on BMI alone as a marker for obesity and associated metabolic health consequences leads to unreliable risk assessment for some individuals. In the current study, while there was a significant difference in the BMI (higher for White/European ancestry) compared to Asian, the mBMIΔ was only marginally higher in the Asia/other ethnicities (Supplementary Fig. 12).

With the large sample size in the discovery cohort (AusDiab, $n = 10,339$) and validation (BHS, $n = 4492$) we stratified individuals into quintiles based on the disparity between mBMI and BMI (mBMIΔ). Despite having a comparable BMIs, the most discordant mBMI groups (Q5 and Q1), displayed distinct metabolic risk profiles. Participants with a mBMI substantially higher than their actual BMI (Q5) presented with a deleterious metabolic profile (i.e., higher triglyceride, HOMA-IR, 2h-PLG and a significantly lower HDL-C) compared to participants with a mBMI substantially less than their BMI (Q1). This was consistent with previous reports in which individuals with an overestimated BMI (based on their metabolism) had higher levels of triglycerides and lower levels of HDL-C compared to those with underestimated BMI[21,33,34]. We also observed that the odds of having a newly diagnosed prevalent T2DM was more than four-fold higher in Q5 compared with Q1, despite Q5 having nearly same average BMI as Q1. Similarly, the risk of 5-year incident T2DM was more than twofold higher in Q5 compared to Q1. These findings have important clinical implications. As mBMI was significantly associated with an increased risk of incident T2DM and incident pre-diabetes, 5 years prior to onset, early pharmacological and lifestyle interventions could be implemented to reduce risk and/or prevent disease progression.

Being overweight or obese based on BMI is a strong risk factor for pre-diabetes and diabetes[37,50,51]. However, recent reports demonstrate varying risk of diabetes across different obesity phenotypes and or metabolic health status[52–54], including a high prevalence of diabetes among normal weight individuals[55,56]. Consequently, relying solely on BMI status to classify obesity is insufficient in providing a comprehensive understanding of an individual's current health condition, and likelihood of experiencing future adverse health outcomes. Here we identified that mBMIΔ associates with T2DM risk independently of BMI and so may be useful in identifying metabolic disturbances, and T2DM risk, in lean individuals. In line to this, a recent study had demonstrated a higher ΔBMI; the difference between metabolome-predicted BMI and actual BMI in the metabolically unhealthy normal weight and metabolically unhealthy obese compared to the metabolically healthy normal weight and MHO, emphasizing the potential of omics-inferred BMI instead of the actual BMI for precise classification of obesity and metabolic health status[34]. The precise phenotyping of metabolic obesity and understanding the difference in metabolically distinct groups may lead to new insights for preventing and treating cardiometabolic diseases. In a sex-stratified analysis, we observed that the odds ratios for the different quintiles of the mBMIΔ were slightly larger in women compared to the men suggesting a stronger association between the metabolic BMI and diabetes (newly diagnosed T2DM) in women. Hormonal differences, and differences in fat distribution[57], metabolism (such as lipids) and lifestyle[24,58] between men and women are likely to contribute for the observed differences.

In the present study, we observed that, mBMIΔ was associated with CVD risk independently of BMI and may explain some of the apparent inconsistencies in associations between BMI and disease outcomes. Consistent with this, a previous study identified, significant differences in cardiovascular events between the different mBMI/BMI groups (higher risk among individuals with a metabolome over-estimated BMI (mBMI>BMI)) compared to those whose mBMI<BMI[21]. While BMI is an independent risk factor for CVD[59,60], not all obese or overweight people show abnormal cardiovascular risk profiles. There is remarkable metabolic heterogeneity in obesity, and hence the risk of CVD[61–63]. Thus, BMI has limited value as a marker of CVD risk. This is highlighted by the absence of BMI in the discriminatory features of the Framingham CVD risk scores[64]. Moreover, a significant portion of obese individuals (31.7%) have been shown to remain free of CVD for life (i.e., metabolically healthy)[65]. Furthermore, a recent debate over the obesity paradox (in which obesity is associated with favourable outcomes and/or improved survival after a CVD event[66–68]) arises partly due to the use of BMI as a single measure to assess CVD risk. The stronger association of mBMI and mBMIΔ with T2D compared to CVD likely reflects the stronger involvement of lipid metabolism, and its dysregulation, in the aetiology of insulin resistance and progression to T2D. In contrast CVD risk likely incorporates other metabolic and inflammatory pathways not covered in this mBMI score.

In this study, we report specific dietary and lifestyle factors independently associated in a strong, dose responsive manner with mBMIΔ, suggesting that targeting these factors might improve an individual's metabolic health. As expected, higher total fruit intake, and dietary fibre consumption were independently associated with a lower mBMIΔ, showing a linear trend across the quintiles of intake. In a recent study, lower fruit and vegetable consumption was reported in participants whose predicted BMI difference (pBMI-BMI) was >5 kg/m$^2$ relative to the normal weight individuals[33]. Indeed, several epidemiological studies have reported an inverse relationship between fruit consumption or dietary fibre and risk of T2D and atherosclerosis[69–72]. We report an inverse association between the level of PA and mBMIΔ but an independent positive association of TV viewing time with mBMIΔ implying that lifestyle habits particularly inadequate exercise and or prolonged sitting time contribute to metabolic risk. Our findings are consistent with prior studies in the AusDiab cohort reporting an inverse association between PA time and 2h-PLG level but not FBG[73] and deleterious associations between TV viewing time and 2h-PLG, WC, BMI, SBP, fasting triglycerides, and HDL-C, but not FBG[74,75]. Taken together, these findings suggest that diet and exercise/sedentary behaviour impact on our metabolism leading to increased risk of impaired glucose tolerance, a key risk factor for T2DM. Indeed, dietary and lifestyle interventions remain important primary prevention strategies for cardiometabolic health management to delay the onset and progression of T2D and CVD[76,77]. mBMI holds potential as a valuable biomarker for tracking the influence of diet and lifestyle on our metabolic health. In a recent study, the implementation of a healthy lifestyle coaching within the Arivale cohort resulted in a significant reduction in mBMI. Importantly, this reduction in mBMI was observed to occur at a faster rate compared to changes in the actual BMI, providing further support for the use of mBMI as an indicator of metabolic health improvements during interventions such as lifestyle coaching programs[34].

The rich lipidomic data, the large sample size and the inclusion of an independent validation cohort as well as the prospective study design of the study cohorts are the major strengths of the present study. However, there are also limitations: (1) As with all such studies we were limited by breadth of the lipidomic profile captured with our platform, although the high proportion of BMI variance explained suggests this is not a major drawback. (2) The lack of some traits such as the 2h-PLG and HbA1c in the BHS validation cohort, however we were able to validate the BMI model and many of the associations in the BHS cohort. (3) Ethnicity of the present study populations was primarily white/European ancestry, and this may limit the generalizability of the findings to other populations. It is likely that normalisation of mBMI will be required for other ethnicities. Indeed, it is

important to acknowledge that the current score was specifically developed and validated for use in adults. However, we acknowledge the importance of addressing the demand for a population-specific score designed specifically for children and adolescents in the future.

In summary, our results demonstrate that mBMI can accurately capture the dysregulation of the plasma lipidomic profile associated with BMI but which is independent of measured BMI. This places mBMI as an important biomarker of metabolic health and a potential tool to monitor dietary and lifestyle interventions to improve metabolic health and reduce cardiometabolic risk. Given the limitations of current lipidomic measurement technologies that hinder clinical applicability, there is a need for a purpose-built clinical platform specifically designed to integrate into healthcare practices[38,78]. Such a platform would provide a reliable means of assessing metabolic health and risk, allowing for informed clinical decision-making.

## Methods

This study used datasets from the AusDiab biobank (project grant APP1101320) approved by the Alfred Human Research Ethics Committee, Melbourne, Australia (project approval number, 41/18) and the BHS cohort (informed consent obtained from all participants, and the study was approved by the University of Western Australia Human Research Ethics Committee [UWA-HREC; approval number, 608/15]). The current study was also approved by UWA HREC (RA/4/1/7894) and the Western Australian Department of Health HREC (RGS03656). Both studies were conducted in accordance with the ethical principles of the Declaration of Helsinki. No participant compensation was provided.

### Australian diabetes, obesity and lifestyle study (AusDiab)

The AusDiab cohort is a national population-based prospective study that was established to study the prevalence and risk factors of diabetes and CVD in an Australian adult population. Some 94.7% of the AusDiab participants were white/European ancestry and 5.3% were Asian/other ancestry (as reported by the participants). The baseline survey was conducted in 1999/2000 with 11,247 participants aged ≥ 25 years randomly selected from the six states and the Northern Territory comprising 42 urban and rural areas of Australia using a stratified cluster sampling method. The detailed description of study population, methods, and response rates of the AusDiab study is found elsewhere[79]. Measurement techniques for clinical lipids including fasting serum total cholesterol, HDL-C, and triglycerides as well as for height, weight, BMI, and other behavioural risk factors have been described previously[80]. We utilized all baseline fasting plasma samples from the AusDiab cohort ($n = 10,339$) (Table 1) after excluding samples from pregnant women ($n = 21$), those with missing data ($n = 277$), technical reasons ($n = 19$) or whose fasting plasma samples were unavailable ($n = 591$). The mean (SD) age was 51.3 (14.3) years with women comprising 5685 (55%) of the cohort. Both sexes were included in this study and sex-stratified analyses were performed whenever necessary.

### The Busselton Health Study (BHS)

We utilized the BHS cohort as a validation cohort. The BHS is a community-based study in the town of Busselton, Western Australia; the participants are of white/European origin. A total of 4492 subjects in the 1994/95 survey of the ongoing epidemiological study were included (Table 1). The mean (SD) age was 50.8 (17.4) years with women constituting 2516 (56%) of the cohort. The details of the study and measurements for HDL-C, LDL-C, triglycerides, total cholesterol, and BMI are described elsewhere[81,82]. The baseline characteristics of study participants are provided in Table 1. Both sexes were included in this study.

### Clinical, lifestyle and dietary data

The demographic and behavioural data collection has been described in detail elsewhere for AusDiab[79,83] and BHS[82]. Fasting plasma cholesterol and lipoprotein concentration including total cholesterol, high density cholesterol, (HDL-C), low density lipoprotein cholesterol (LDL-C) and triglycerides, fasting plasma glucose (FPG) and 2 h post load glucose (2h-PLG) were measured using standard protocols[84]. Methods for assessment of dietary intake, PA time and TV viewing time are provided in the Supplementary Note 2.

### Clinical endpoints and outcome ascertainment

Diabetes status was ascertained using the American Diabetes Association criteria (FBG > = 7.0 mmol/L or 2h-PLG > = 11.1 mmol/L after a 75-g oral glucose load)[85]. In the AusDiab cohort, both a newly diagnosed prevalent T2DM ($n = 395/7733$ NGT) and 5-year incident ($n = 218/5354$ controls) were included. Participants with newly diagnosed prevalent T2DM are those not receiving pharmacological treatment for diabetes, nor previously diagnosed with diabetes, and who had FBG or 2h-PLG measurements over the diabetes cut-off range. Participants were classified as having IFG, if FBG was 6.1–6.9 mmoL/L and 2h-PLG was <7.8 mmol/L and IGT if FBG < 7 and 2h-PLG is 7.8 – 11.0 mmol/L. The detailed diagnostic criteria for the presence of diabetes and pre-diabetes can be found elsewhere[86]. In the AusDiab cohort, some 577 prevalent CVD (history of heart attack and stroke combined) and 414 major CVEs were recorded over 10 years of follow-up. The major CVEs included IHD (angina pectoris, myocardial infarction, coronary artery bypass grafting and percutaneous transluminal coronary angioplasty), cerebrovascular diseases (intracerebral haemorrhage, cerebral infarction and stroke). The CVE outcomes are defined based on the international classification of diseases (ICD) codes and ascertained through linkage to the National Death Index and medical records. The detailed baseline characteristics of the AusDiab participants in the disease and control groups can be found in Supplementary Table 1. In the BHS cohort, there were 238 prevalent CVD cases and 4254 controls ascertained through health linkage data at baseline and 284 IHD events (including myocardial infarction, angina, coronary artery bypass grafting and percutaneous transluminal coronary angioplasty) recorded over 10 years follow up (Fig. 1, Supplementary Table 2). The baseline characteristics of those who had an event and those who hadn't are summarized in Supplementary Table 2.

### Lipidomics

A butanol/methanol extraction method[30] was used to extract lipids from human plasma. Briefly, 10 μL of plasma was mixed with 100 μL of a 1-butanol and methanol (1:1 v/v) solution containing 5 mM ammonium formate and the relevant internal standards (Supplementary Data 4). The resulting mix was vortexed (10 seconds) and sonicated (60 min, 25 °C) in a sonic water bath. Immediately after sonication, the mix was centrifuged (16,000 × g, 10 min, 20 °C). The supernatant was transferred into samples tubes containing 0.2 ml glass inserts and Teflon seals. The extracts were stored at −80 °C until analysed by liquid chromatography tandem mass spectrometry (LC-MS/MS).

Targeted lipidomic analysis was performed using liquid chromatography electrospray ionization tandem mass spectrometry (LC-ESI-MS/MS). An Agilent 6490 triple quadrupole (QQQ) mass spectrometer [(Agilent 1290 series HPLC system and a ZORBAX eclipse plus C18 column (2.1 × 100 mm 1.8 μm, Agilent)]) in positive ion mode was used [details of the method and chromatography gradient have been described previously[23]]. Compared to our earlier study, we modified the methodology to enable a dual column setup (while one column runs a sample, the other is equilibrated) to increase throughput[23] for the AusDiab. In brief, the temperature was reduced to 45°c from 60°c with modifications to the chromatography to enable similar level of separation. Starting at 15% solvent B and increasing to 50% B over 2.5 min, then quickly ramping to 57% B for 0.1 min. For 6.4 min, %B was increased to 70%, then increased to 93% over 0.1 min and increased to 96% over 1.9 min. The gradient was quickly ramped up to 100% B for 0.1 min and held at 100% B for a further 0.9 min. This is a total run time

of 12 min. The column is then brought back down to 15% B for 0.2 min and held for another 0.7 min prior to switching to the alternate column for running the next sample. The column that is being equilibrated is run as follows: 0.9 min of 15% B, 0.1 min increase to 100% B and held for 5 min, decreasing back to 15% B over 0.1 min and held until it is switched for the next sample. We used a 1-μL injection per sample with the following mass spectrometer conditions were used: gas temperature, 150 °C; gas flow rate, 17 L/min; nebuliser, 20 psi; sheath gas temperature, 200 °C; capillary voltage, 3500 V; and sheath gas flow, 10 L/min. Given the large sample size, samples were run across several batches, as described above. The LC-MS/MS conditions and settings with the respective MRM transitions for each lipid ($n = 747$) can be found in Supplementary Data 4. For the BHS, lipidomic profiling was performed using the standardised methodology as in the AusDiab except in BHS thermostat set at 60 °C and single column rather than dual column was used[23,35]. Overall, 596 lipid species were quantified; 575 of which were common to AusDiab cohort.

## Lipidomic nomenclature

The lipid nomenclature employed in this study follows the established guidelines set by the Lipid Maps Consortium and incorporates additional recommendations made by experts in the field[87–89]. Glycerophospholipids, which typically consist of two fatty acid chains, are represented as the sum composition of carbon atoms and double bonds (e.g., PC(38:6)) when detailed characterization is not available. In cases where the acyl chains have been identified but their positions are unknown, an underscore is used to indicate this uncertainty (e.g., PC(38:6) is modified to PC(16:0_22:6)). If the positions of the acyl chains are known, they are separated by a forward slash (/) and listed in order of the sn1 and sn2 positions (e.g., PC(16:0_22:6) is changed to PC(16:0/22:6)). This naming convention extends to other lipid classes and subclasses as well. In instances where chromatographic separation is incomplete, but species are partially characterized, labels such as (a) or (b) are used to denote the elution order, as exemplified by PC(P-17:0/20:4) (a) and (b). Similarly, glycerolipids are named as the sum composition of carbon atoms and double bonds with the fatty acyl defined by the neutral loss (NL) fragmentation in the mass spectrometer also annotated. For example, TG(52:2)[NL-18:0] is the notation for a triglyceride (TG) molecule where 52 represent the total number of carbon atoms and 2 is the number of double bonds. The [NL-18:0] refers to the presence of an 18:0 acyl chain within the structure.

## Data pre-processing and quality

To ensure the robustness of the lipid measures, we employed state-of-the-art lipidomic profiling techniques that are designed to capture a wide range of lipid species, including those with lower abundances. Integration of the chromatograms for the corresponding lipid species was performed using Agilent Mass Hunter version 8.0. Relative quantification of lipid species was determined by comparing the peak areas of each lipid in each patient sample with the relevant internal standard (Supplementary Data 4). A median centring approach was carried out to correct for batch effect i.e. remove technical batch variation using PQC samples[90] in both AusDiab and BHS. Briefly, the lipidomic data in each batch consisting about 485 samples was aligned to the median value in pooled PQC samples included in each run. More than 90% of the lipid species were measured with a coefficient of variation <20% (based on PQC, samples). TQC samples every 20 samples were included in the runs allowing for the assessment of technical variation that arises from the mass spectrometer. NIST 1950 reference plasma sample (Gaithersburg, MD, USA) for every 20 samples were included to facilitate future alignment with other studies. While the overall median %CV for TQC, PQC and NIST were 10.9, 10.7 and 10.7 respectively (Supplementary Data 5), LPC species are among the many lipid species measured with low variability (median PQC %CV = 8.8) (Supplementary Data 5). Ceramides, as expected had a relatively higher median %CV

(13.1) (Supplementary Data 5). The lipids selected into the LASSO model and top 50 lipids in the ridge (chiefly sphingomyelin and phospholipids species) (Supplementary Data 3) had lower CVs; a median %CV of 10.5 and 7.4 respectively (Supplementary Data 5). Only technical outliers ($n = 19$ samples) were excluded from the downstream analysis for the AusDiab. In this study, we utilised lipid species ($n = 575$) spanning across the sphingolipid, glycerophospholipid and glycerolipid categories that were common in both study cohorts (AusDiab and the BHS). These were used for model development.

## Predictive modelling

Lipidomic data was log10 transformed, mean centred and scaled to unit SD prior to statistical analysis. A ridge regression model including age, sex and the lipidome (comprising 575 lipid species common to the AusDiab and the BHS cohorts) was employed to determine a predicted BMI (pBMI). In addition, Elastic-Net and least absolute shrinkage and selection operator (LASSO) models were also developed to predict BMI. A 10-fold cross validation was employed for the generation pBMI scores in the AusDiab (i.e., models trained on the 9/10th and used to predict BMI in holdout 1/10th of the cohort). The lambda parameter was optimized using cv.glmnet R package, minimizing the MSE, lambda range restricted between 0.2 and -4.0 on log10 scale. A metabolic BMI (mBMI) was derived from the pBMI scores as follows: mBMI = BMI + (pBMI – pBMI value on the line of best fit between pBMI and BMI). BMI prediction models were cross-validated in the AusDiab cohort and used to predict BMI in the BHS cohort. The mBMI in BHS was calculated using coefficients and line of best fit from the original model developed in AusDiab. The mBMI values were also calculated for the National Institutes of Standards Technology standard reference material (NIST 1950) QC samples using a value of 26 as the measured BMI. The %CV of the NIST mBMI scores were calculated after excluding technical outliers. Further to the optimized models, we established a LASSO framework to generate an array of models ($n = 120$ different models) with the respective lambda value between 0.2 and -4.0 on log10 scale or the number of features selected into the model ranging from all lipid species to null.

## Statistical analysis

The difference between the mBMI and the BMI, termed the 'mBMIΔ', was used to stratify individuals into quintiles. Z-score values for cardiometabolic traits were calculated as follows [$(z = x - mean(x))/SD(x)$] to allow better comparison across groups. A linear regression analysis was performed between cardiometabolic traits (outcome) and the quintiles of mBMIΔ (as a predictor). The association of cardiometabolic risk factors with metabolic discordant groups (Q5 relative to Q1) were evaluated by using logistic regression adjusting for age, sex and BMI and other appropriate covariates. Linear regression models were used to examine the association of mBMIΔ or BMI with the plasma lipidomic profile adjusting for the appropriate covariates and correcting p-values for multiple comparison using the Benjamini-Hochberg procedure[91]. The Akaike information criteria (AIC) was used to assess the relative quality of individuals models with and without mBMIΔ.

A logistic regression model was used to assess the relationship between the mBMIΔ or quintiles of mBMIΔ and pre-diabetes or T2DM (both prevalent and the 5-year incident cases) adjusting for age, sex and BMI or these covariates plus clinical lipids, familial history of diabetes, and smoking status. Further, we examined the association of mBMIΔ with the prevalent CVD and incident CVEs adjusted for age, sex, BMI, smoking and diabetes history or these covariates plus clinical lipids. The adjustment for clinical lipids was performed as a sensitivity analysis, motivated by the aim of evaluating the additional value of lipidomics in predicting metabolic status beyond / independent of traditional clinical lipid measures. Cox regression models were fitted to compute hazard ratios (HRs) associated with CVEs that occurred

during the 10 years follow up using age as the time scale using coxph() function in the survival package while logistic regression was used for prevalent cases.

Multivariable linear regression was performed to assess the associations between dietary components such as total fruit intake or lifestyle habits such as total leisure PA time and TV viewing time (as predictor variables) and mBMIΔ (as a continuous outcome variable). We created two different models: model 1 (age, sex and BMI adjusted) and model 2 additionally adjusted for potential confounders such as intake of daily total energy, total alcohol, total fat, carbohydrate, sugar, processed meat, red meat, tinned fish, total fibre, fruit intake and total protein as continuous variables and smoking, baseline diabetes status and history of cardiovascular disease, and educational level as dichotomous variables. STATA v15 (StataCorp LP, Inc., Texas, USA) or R (version 3.6.1) were used to analyse the data as necessary.

## Reporting summary
Further information on research design is available in the Nature Portfolio Reporting Summary linked to this article.

## Data availability
Because of the participant consent obtained as part of the recruitment process for the Australian Diabetes, Obesity and Lifestyle Study, it is not possible to make data publicly available (including the individual deidentified data). Individual-level data are available for analyses that do not conflict with ongoing studies, through application to the study lead Professor Jonathan Shaw and the AusDiab Study Committee (Email: Jonathan.Shaw@baker.edu.au). The timeframe for response to such requests is within two months.

Individual-level data for the Busselton Health Study are available under restricted access for analyses that do not conflict with ongoing studies; access is available through application to the Busselton Population Medical Research Institute (http://bpmri.org.au/research/database-access.html). Responses will be provided within 2 months.

The complete summary statistics for the Australian Diabetes, Obesity and Lifestyle Study and the Busselton Health Study are provided in the manuscript and Supplementary files. Source data are provided with this paper.

## Code availability
All software and bioinformatic tools are publicly available including R packages (https://cran.r-project.org/package=glmnet, https://cran.r-project.org/package=ggplot2, https://cran.r-project.org/package=ggExtra, https://cran.r-project.org/package=survival).

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

## Acknowledgements

This research was supported by the National Health and Medical Research Council of Australia (Project grant APP1101320 to JES, DJM and PJM). This work was also supported in part by the Victorian Government's Operational Infrastructure Support Program. The AusDiab study, initiated and coordinated by the International Diabetes Institute, and subsequently coordinated by the Baker Heart and Diabetes Institute, gratefully acknowledges the support and assistance given by: A Allman, B Atkins, S Bennett, A Bonney, S Chadban, M de Courten, M Dalton, D Dunstan, T Dwyer, H Jahangir, D Jolley, D McCarty, A Meehan, N Meinig, S Murray, K O'Dea, K Polkinghorne, P Phillips, C Reid, A Stewart, R Tapp, H Taylor, T Welborn, T Whalen, F Wilson, P Zimmet and all the study participants. Also, for funding or logistical support, we are grateful to: National Health and Medical Research Council (NHMRC grant 233200), Australian Government Department of Health and Ageing. Abbott Australasia Pty Ltd, Alphapharm Pty Ltd, AstraZeneca, Bristol-Myers Squibb, City Health Centre-Diabetes Service-Canberra, Department of Health and Community Services—Northern Territory, Department of Health and Human Services—Tasmania, Department of Health—New South Wales, Department of Health – Western Australia, Department of Health—South Australia, Department of Human Services—Victoria, Diabetes Australia, Diabetes Australia Northern Territory, Eli Lilly Australia, Estate of the Late Edward Wilson, GlaxoSmithKline, Jack Brockhoff Foundation, Janssen-Cilag,, Kidney Health Australia, Marian & FH Flack Trust, Menzies Research Institute, Merck Sharp & Dohme, Novartis Pharmaceuticals, Novo Nordisk Pharmaceuticals, Pfizer Pty Ltd, Pratt Foundation, Queensland Health, Roche Diagnostics Australia, Royal Prince Alfred Hospital, Sydney, Sanofi Aventis, sanofi-synthelabo, and the Victorian Government's OIS Program. JES, DJM and PJM are supported by Investigator grants from the National Health and Medical Research Council of Australia. HBB was supported by the Baker institute and Monash University Scholarships. The authors wish to thank the staff at the Western Australian Data Linkage Branch and Death Registrations and Hospital Morbidity Data Collection for the provision of linked health data for the BHS. The 1994/95 BHS was supported by a grant from the Health Promotion Foundation of Western Australia, and the authors acknowledge the generous support for the 1994/1995 BHS follow-up from Western Australia and the Great Wine Estates of the Margaret River region of Western Australia. Support from the Royal Perth Hospital Medical Research Foundation is also gratefully acknowledged. The funders had no role in study design, data collection and analysis, decision to publish, or preparation of the manuscript.

## Author contributions

H.B.B. extracted plasma samples, performed LC-MS/MS analysis, analysed the data and wrote the manuscript. G.O., C.G., T.W. & T.M. provided statistical support and also reviewed the paper. C.G. and K.H. developed LC-MS/MS methods and provided support for the LC-MS/MS analysis and statistical analysis. M.C. developed extraction protocols and extracted plasma samples. N.M. supported the LC-MS/MS experiment and data pre-processing and analysis. G.W., J.o.H., J.e.H. G.C., J.B.e, J.B.l and E.M. were involved in review & editing. J.E.S. and D.J.M., coordinated the AusDiab data, interpreted results and revised the manuscript. P.J.M. oversaw this work and revised the manuscript. P.J.M. and D.J.M. are the guarantors of this work and shall take the responsibility for the full access and integrity of the data. All authors have approved the final version of the manuscript.

## Competing interests

The authors declare no competing interests.
