## [Peer Review File · Nature Communications]

Metabolic phenotyping of BMI to characterize cardiometabolic risk: evidence from large population-based cohortsREVIEWER COMMENTS

Reviewer #1 (Remarks to the Author):

The manuscript presents a methodology to calculate metabolic BMI that is a better predictor of cardiometabolic risk than traditional BMI, using a comprehensive lipid profile in human plasma. The concept as such is not fully novel, as similar approaches have been reported earlier, that is, metabolome-defined obesity and multiomic signatures of BMI have recently been reported. The study highlights that BMI alone is not a sufficient measure of future risk of cardiometabolic diseases and by utilizing lipidomic/metabolic markers, a more accurate measure of individuals metabolic status can be obtained.

The strength of the study is the very big longitudinal clinical cohort, and the results were validated in an additional, large validation cohort, with both cohorts having comprehensive lipidomics measurements available. Overall, the study is well done, with state-of-the-art methodologies and the manuscript is well written. One of the main issues is that as the model would require measurements of 100-200 lipids to be accurate, requiring also more complex, methodology than is currently available in clinical laboratories, this methodology is not well suited into clinical practice. This should be discussed in the manuscript. Also, there is not any data that would compare the developed model against available models that utilize not only BMI but other clinical measurements, such as clinical lipids and measures of glucose control. It would be important to see this comparison, to see if a very comprehensive lipid panel, that is potentially difficult to bring into clinical practice, would be useful in clinical decision making.

The clinical values (clinical lipids, 2h-PLG, HbA1c, HOMA-IR etc) show a strong association with the mBMI Δ . Would it be possible to calculate mBMI Δ with these values, and how would that compare with the current mBMI Δ ?

It is not clear what lipids are in the model that could be done with 100-200 lipids. It would be good to mark those lipids, e.g. in the supplementary tables. Also, some discussion of the type of lipids could be useful. Now several of the lipids that most strongly associate with mBMI Δ are lipids that have a low abundance in human serum., such as very long chain LPCs

and some ceramides. How robustly are these measured in the samples?

The adjustment for confounding parameters is not sufficiently justified. Age and sex are known to impact the metabolic profiles, however, it would be useful to show how the age and sex associate with the lipids measured? How would the model perform separately for males and females? Some of the models were additionally adjusted for clinical lipids, but it is unclear what was the reasoning behind this adjustment. How do the clinical lipids and lipidomics data correlate? There are sex differences in the prevalence and effect of cardiovascular and diabetes risk factors, so rather than merely adjusting for sex, the males and females should also be investigated separately.

Is the current model, using mBMI Δ for estimating the risk of future T2DM and pre-diabetes and risk of CV better than the available models for estimation of the risk? Now only the BMI has been used in these estimations, but there are models that include other clinical variables and that are more accurate than BMI alone.

Minor comments:

- A recent paper from K. Watanabe et al (Multiomic signatures of body mass index identify heterogeneous health phenotypes and responses to a lifestyle intervention. Nat Med 29, 996–1008 (2023)) should also be covered in the Introduction and discussion.
- Figure 1. Please explain all the abbreviations (e.g. NGT, IFG, IGT etc)
- Please explain the lipid naming in the manuscript and supplements, e.g. TG(52:1)[NL-18:0].

Reviewer #2 (Remarks to the Author):

The paper entitled “Metabolic Phenotyping of BMI to Characterize Cardiometabolic Risk: Evidence from Large Population-Based Cohorts” by Beyene et al, presented a new metric that can assess obesity better than the traditional BMI and predict long-term metabolic complications. The authors were able to create and introduce a new measure for obesity (metabolic health) that they termed metabolic BMI (mBMI). They accomplished the task by utilizing targeted lipidomics that was performed on two large independent Australian cohorts; the discovery cohort is AusDiab (Australian Diabetes, Obesity and Lifestyle Study) and the validation cohort is BHS (Busselton Health Study). By calculating the difference

between metabolic BMI and measured BMI (mBMI delta; “mBMI Δ ”), they were able to categorize their population into quintiles and identify individuals with a similar measured BMI but differ in their metabolic health profiles. When they compared the two most discordant groups (Q1) and (Q5), they illustrated that participants in the top quintile of mBMI Δ (Q5) are four times more likely to be newly diagnosed with T2DM, two times more likely to develop T2DM over a five year follow up period and had higher odds of cardiovascular disease relative to those in the bottom quintile (Q1). They further show an association between diet and exercise with the mBMI Δ score suggesting potential utilization of their new score (mBMI) in designing lifestyle interventions. The author concluded that mBMI can play an important role as a biomarker of metabolic health that is independent from measured BMI.

Overall, this longitudinal study is well designed, and the paper is well written. The findings within are important and potentially impactful especially since they were validated via several methods. The manuscript would be recommended for publications after addressing the following points:

- It would be of great value if these findings complemented with fat body distribution data to indicate the association between metabolic BMI and the different fat depots in the body. This will further validate the utility of this metabolic BMI. The authors need to attempt to utilize the waist-circumference as an additional supportive association!
- It would be great if the analysis for the metabolic score be developed based on common lipid markers such as LDL, TC, HDL, vLDL and TG to see how well these routine lipid markers fair with the calculated metabolic BMI score. The reason for this is to better understand if these commonly measured markers would be useful and comparable to predict the mBMI or not. A statement on this issue may be of value.
- This reviewer understands that the metabolomic profiling was done at baseline. However, it would enrich the paper if some samples from the same individuals were also measured longitudinally (at future time). Metabolomics change with time! Such exercise may reassure us that the observed mBMI was indeed maintained or changed especially in relation to the next point raised!!

- The term metabolically healthy obese refers to a timepoint. The idea that almost all metabolically healthy obese convert with time to metabolically unhealthy obese need to be discussed in line with our work that needs to be cited and discussed (PMID: 33827711, PMID: 35109943).

- The authors mention that both cohorts are primarily of White/European ancestry. Can they add the exact numbers/percentages in Table 1? That would be easier for the reader than going back to previous publications. Would it be possible to adjust their analysis for ethnicity?

- Can the author discuss the main differences between the mBMI score that they are proposing and the metabolomic signatures proposed by others (e.g. Cirulli et al.2019) beyond explaining variation in BMI? Specifically, how sensitive is the mBMI compared to metabolomic signatures in predicting metabolic complications?

- Why is LDL-C not included in the analysis? Can the authors comment on that?

- How does waist circumference (as a measure of central obesity) associate with the newly-introduced score (mBMI and mBMI Δ)?

- Why was family history of diabetes adjusted for in the analysis for CVD risk (Table 2) but not in prevalent and incident T2DM risk (Supplementary Table 7)?

- Using ridge regression model, the pBMI can explain 60.4% of the BMI variation in the AusDiab cohort and 40% of the BMI variation in the BHS cohort. Can the author suggest possible approaches to improve these numbers?

- The authors should be more concise in their discussion on dietary and lifestyle factors. While they clearly show the association with mBMI, their title in the discussion “mBMI can be modified by dietary and lifestyle factors” gives the false impression that they actually conducted an intervention and showed that they can change mBMI by modifying diet or improving physical activity.

- Childhood obesity is currently classified using WHO growth charts after measuring BMI-for-age, can the authors add a paragraph in their discussion on whether their new score can be adapted for children and adolescents?

- The authors mention that compared to direct methods for measuring adiposity, BMI is an inexpensive, simple, and easily interpretable metric. Can they add a comparison in their discussion between these methods (e.g. computed tomography and dual energy x-ray absorptiometry) and performing targeted lipidomics from cost-effectiveness and clinical application?

REVIEWER COMMENTS

Reviewer #1 (Remarks to the Author):

The manuscript presents a methodology to calculate metabolic BMI that is a better predictor of cardiometabolic risk than traditional BMI, using a comprehensive lipid profile in human plasma. The concept as such is not fully novel, as similar approaches have been reported earlier, that is, metabolome-defined obesity and multiomic signatures of BMI have recently been reported. The study highlights that BMI alone is not a sufficient measure of future risk of cardiometabolic diseases and by utilizing lipidomic/metabolic markers, a more accurate measure of individuals metabolic status can be obtained.

Response: Thank you for bringing up the observation that similar approaches, such as metabolome-defined obesity and multiomic signatures of BMI, have been reported earlier. It is indeed important to acknowledge and build upon existing research in a field. These early studies have identified mBMI scores that capture residual risk of a range of cardiometabolic outcomes. However, the signal being captured by mBMI scores has not been clearly defined nor has the relationship with disease outcomes been adequately quantified. Aiming to addressing these issues, and improve the statistical power, the current work focused on metabolic BMI calculation using a comprehensive lipid profile in two large datasets, including a validation cohort. We quantified the mBMI-disease relationships and how diet and lifestyle factors relate to mBMI. We further have been able to define the nature of the additional information obtained from the mBMI, in terms of the association of mBMI Δ with the lipid species and highlight this alignment to the BMI associations. This is an important step forward in our understanding of mBMI and related metabolic scores.

The strength of the study is the very large longitudinal clinical cohort, and the results were validated in an additional, large validation cohort, with both cohorts having comprehensive lipidomics measurements available. Overall, the study is well done, with state-of-the-art methodologies and the manuscript is well written. One of the main issues is that as the model would require measurements of 100-200 lipids to be accurate, requiring also more complex, methodology than is currently available in clinical laboratories, this methodology is not well suited into clinical practice. This should be discussed in the manuscript.

Response: Thank you for recognizing the strengths of our work, particularly the large longitudinal clinical cohort and the additional validation cohort that enhance the reliability and generalizability of the findings. It is indeed an important consideration, regarding the concern raised about the applicability of the methodology in clinical practice. While the current methodology may not be immediately suited for routine clinical practice, it does provide valuable insights into the potential for incorporating comprehensive lipidomics measurements in metabolic risk assessment. As the technology continues to advance, it is possible that lipidomics may become more accessible and practical in clinical laboratories. We have recently published an article entitled "Clinical lipidomics: realizing the potential of lipid profiling" discussing current state of clinical lipidomics, specifically examining the analysis of lipid profiles in clinical samples, prospective design of clinical platforms, and the potential pathways for implementing clinical lipidomics in routine clinical practice [ref 38]. The authors envision the future development of purpose built specialized clinical platforms that are specifically designed for integration into healthcare settings to help healthcare providers assess, diagnose, and monitor disease risk more effectively. Indeed, charting the path forward for clinical lipidomics will require a collaborative roadmap

and workflow harmonization across laboratories [ref 80]. We have now included a brief discussion around this in the manuscript (Page 19, line 494-499 and page 23, line 678-682).

Also, there is not any data that would compare the developed model against available models that utilize not only BMI but other clinical measurements, such as clinical lipids and measures of glucose control. It would be important to see this comparison, to see if a very comprehensive lipid panel, that is potentially difficult to bring into clinical practice, would be useful in clinical decision making.

Response: The reviewer raises an important point about comparing the developed mBMI model with existing models that incorporate clinical measurements such as clinical lipids and measures of glucose control. However, we would like to emphasize that the mBMI score is primarily meant to serve as a biomarker for an individual's metabolic health status rather than improving disease prediction models (page 20, lines 503-508). Moreover, the score is useful to explain heterogeneity in the metabolic states/risk in people with comparable BMIs and identify those who might benefit most from diet and lifestyle intervention.

The clinical values (clinical lipids, 2h-PLG, HbA1c, HOMA-IR etc) show a strong association with the mBMI Δ . Would it be possible to calculate mBMI Δ with these values, and how would that compare with the current mBMI Δ ?

Response: We have calculated mBMI Δ using models developed with 1) only clinical lipids (total cholesterol, HDL-C and triglycerides or 2) cardiometabolic risk factors (CMRs, clinical lipids plus HBA1C, FBG, 2h-PLG, HOMA-IR SBP and DBP) and compared the results to the current mBMI Δ . The findings revealed that when using only clinical lipid measures, the resulting model explained 15.6% of the variation in BMI within the AusDiab cohort. Upon incorporating all cardiometabolic risk factors (clinical lipids, HBA1C, FBG, 2h-PLG, HOMA-IR SBP and DBP), the model explained, 31.6% variation in BMI (page, 6, line 184-188, and Supp Figure 3). In comparison the lipidomic model explained 60.4% of the variation in BMI (Figure 2 panel A).

The mBMI Δ calculated from the model using only clinical lipids exhibited a strong correlation ($r=0.68$) with the mBMI Δ calculated using the model incorporating CMRs. However, it is important to note that the correlation between mBMI Δ derived from the clinical lipid values or the CMRs and the score calculated using lipidomic data (the current mBMI Δ) were weaker, ($r = 0.3$ and 0.34 respectively) (Supp Figure 10A). This suggests that the lipidomic data captures independent information that is not fully accounted for by the clinical lipid values or the CMRs alone. Therefore, incorporating lipidomic data can provide additional insights into an individual's metabolic status and contribute to a more comprehensive assessment of their cardiometabolic health. We also examined how the mBMI Δ derived from clinical lipids or CMRs related with disease outcomes and how these compared with the current mBMI Δ . Although, the BMI prediction performance was low when using just clinical lipids or CMRs (Supp Figure 3), the mBMI Δ calculated from clinical lipid values performed as well as the mBMI Δ of the lipidome model in predicting T2DM. As expected, mBMI Δ calculated from the CMRs model performed better than the lipidomic model at prediction of prevalent and incident T2DM, as the diagnostic criteria for diabetes are included in the model. In contrast the mBMI Δ derived from model with clinical lipids or CMRs did not predict cardiovascular disease, demonstrating the limitations of these

models (Supp Figure 10B-10D). We have included this additional information to the manuscript (page 17-18, line 432-445).

It is not clear what lipids are in the model that could be done with 100-200 lipids. It would be good to mark those lipids, e.g., in the supplementary tables. Also, some discussion of the type of lipids could be useful. Now several of the lipids that most strongly associate with mBMI Δ are lipids that have a low abundance in human serum., such as very long chain LPCs and some ceramides. How robustly are these measured in the samples?

Response: Thank you for bringing up these important points. The weighting of the individual lipid species in both the Ridge and LASSO models are shown in Figure 3 (Panel D and E) (page 10). We have also included a table with these values (Supp Table 6). We note the prominent role of sphingomyelin species, such as SM(d18:2/14:0), in the models as illustrated in Figure 3D and 3E. Furthermore, these figures highlight the lipid species that make the greatest contribution to the Ridge and LASSO models with species of sphingomyelin and several phospholipid classes playing a prominent role. We have commented on this in the results section of the manuscript (page 8-9, lines 258-262) and discussion (Page 21, lines 571-575).

Regarding the robustness of the lipid measures, particularly those contributing to the models, we employed state-of-the-art lipidomic profiling techniques that are designed to capture a wide range of lipid species, including those with lower abundances. Although it is acknowledged that some lipids, with low abundance, can present challenges in measurement, our study implemented robust protocols and quality control measures to ensure reliable quantification of these lipids in plasma. We employ different QC materials (PQC, TQC and NIST-1950) samples to monitor the quality our lipid measurements. Over 90% of the lipids were measured with %CV<20. We have included the summary data of the QC materials now (Supp Table 17).

The overall median %CV for TQC, PQC and NIST were 10.9, 10.7 and 10.7 respectively (Supp table 17). Of note, the LPC species are among the many lipid species we measure with low variability (median %CV =8.8) (Supp Table 17). Ceramides, as expected had a relatively higher median %CV of 13.1 based on PQC samples (Supp Table 17). The lipids selected into the LASSO model had a median %CV of 10.5 (lower than the overall) and the top 50 lipids in ridge (ranked based on the absolute value of weightings) had a median %CV of 7.4. We have updated the manuscript (methods section accordingly) (page 25-26 lines 791-800).

The adjustment for confounding parameters is not sufficiently justified. Age and sex are known to impact the metabolic profiles; however, it would be useful to show how the age and sex associate with the lipids measured? How would the model perform separately for males and females? Some of the models were additionally adjusted for clinical lipids, but it is unclear what was the reasoning behind this adjustment. How do the clinical lipids and lipidomics data correlate? There are sex differences in the prevalence and effect of cardiovascular and diabetes risk factors, so rather than merely adjusting for sex, the males and females should also be investigated separately.

Response: Thank you for raising important points regarding the adjustment for confounding parameters in our study. We acknowledge the impact of age and sex on metabolic profiles and their potential influence on the associations observed between lipids and other variables. Prior to this study, we have published results on the

comprehensive analysis on the association of the measured lipids with age and sex employing the same dataset identifying, strong age and sex associations with plasma lipidome [ref 24] (referenced on page 14, line 359-360). Regarding the performance of the models developed separately for males and females, we agree that investigating sex-specific effects is crucial given the known sex differences in the prevalence and impact of cardiometabolic disease. We conducted subgroup analyses separately for males and females including prediction of BMI. The metabolic data explained 67% of BMI variation in women and 55% in men (page, 6, line 180-181, Supp Figure 2). Within these models, the mBMI Δ showed a slightly stronger association with diabetes (Supp Figure 6) and CVD (Supp Figure 9) in women than men, although this difference was marginal. We have now updated the manuscript by highlighting some of the sex effect in the mBMI Δ -disease associations (Results section, Page 14, line 361-364, and page 17, line 413-415) and on page 22, line 620-624).

The adjustment for clinical lipids was performed as a sensitivity analysis to evaluate the additional value of lipidomics in predicting metabolic status beyond / independent of traditional clinical lipid measures. We believe, that the lipidome captures a substantial biological signal which is independent of the clinical lipids as evidenced by the small effect of clinical lipids in these models. The rationale behind the clinical lipid adjustment is now clarified and discussed in the manuscript (page 26, line 840-842).

Is the current model, using mBMI Δ for estimating the risk of future T2DM and pre-diabetes and risk of CV better than the available models for estimation of the risk? Now only the BMI has been used in these estimations, but there are models that include other clinical variables and that are more accurate than BMI alone.

Response: Yes, we agree with the reviewer, mBMI does not outperform existing risk scores for T2DM or CVD. However, this is not the proposed role for the mBMI score. The mBMI score is presented as a measure of the metabolic dysregulation associated with obesity. While this dysregulation is associated with several cardiometabolic diseases, it does not capture all the risk factors for these diseases, such as age, blood pressure, fasting glucose etc. Rather the mBMI presents an opportunity to better identify those individuals who are metabolically unhealthy independently of their BMI and likewise those metabolically healthy individuals who may be overweight/obese based upon BMI alone. By focusing on a metabolic health assessment, the mBMI score can provide valuable insights into an individual's metabolic risk profile and potential areas for targeted interventions (such as diet and lifestyle modification). We have now clarified this within the discussion (Page 20, lines 503-508).

Minor comments:

•A recent paper from K. Watanabe et al (Multiomic signatures of body mass index identifies heterogeneous health phenotypes and responses to a lifestyle intervention. Nat Med 29, 996–1008 (2023)) should also be covered in the Introduction and discussion.

Response: We appreciate the mention of the recent paper by K. Watanabe et al titled "Multiomic signatures of body mass index identify heterogeneous health phenotypes and responses to a lifestyle intervention" published in Nature Medicine (2023). This study provides significant contributions to our understanding of the complex relationship between BMI and health outcomes with multiomic data and the exploration of diverse health phenotypes. We have now incorporated a discussion of their findings in our current

manuscript on page 3 (page 3, lines 93-96) and pages 20, line 544-547 and 22, line 614-618 (discussion section).

- Figure 1. Please explain all the abbreviations (e.g., NGT, IFG, IGT etc)

Response: In Figure 1 (page 4), we have now included the detailed explanations for all abbreviations used to ensure clarity. This includes definitions for NGT (Normal Glucose Tolerance), IFG (Impaired Fasting Glucose), IGT (Impaired Glucose Tolerance), and any other relevant abbreviations used in the figure.

- Please explain the lipid naming in the manuscript and supplements, e.g. TG(52:1)[NL-18:0].

Response: The lipid naming convention used in the manuscript and supplements follows the guidelines established by the Lipid Maps Consortium. We have now updated the manuscript by including a nomenclature section in the manuscript for convenience (page 25, lines 766-781).

Reviewer #2 (Remarks to the Author):

The paper entitled “Metabolic Phenotyping of BMI to Characterize Cardiometabolic Risk: Evidence from Large Population-Based Cohorts” by Beyene et al, presented a new metric that can assess obesity better than the traditional BMI and predict long-term metabolic complications. The authors were able to create and introduce a new measure for obesity (metabolic health) that they termed metabolic BMI (mBMI). They accomplished the task by utilizing targeted lipidomics that was performed on two large independent Australian cohorts; the discovery cohort is AusDiab (Australian Diabetes, Obesity and Lifestyle Study) and the validation cohort is BHS (Busselton Health Study). By calculating the difference between metabolic BMI and measured BMI (mBMI delta; “mBMI Δ ”), they were able to categorize their population into quintiles and identify individuals with a similar measured BMI but differ in their metabolic health profiles. When they compared the two most discordant groups (Q1) and (Q5), they illustrated that participants in the top quintile of mBMI Δ (Q5) are four times more likely to be newly diagnosed with T2DM, two times more likely to develop T2DM over a five year follow up period and had higher odds of cardiovascular disease relative to those in the bottom quintile (Q1). They further show an association between diet and exercise with the mBMI Δ score suggesting potential utilization of their new score (mBMI) in designing lifestyle interventions. The author concluded that mBMI can play an important role as a biomarker of metabolic health that is independent from measured BMI. Overall, this longitudinal study is well designed, and the paper is well written. The findings within are important and potentially impactful especially since they were validated via several methods. The manuscript would be recommended for publications after addressing the following points: - It would be of great value if these findings complemented with fat body distribution data to indicate the association between metabolic BMI and the different fat depots in the body. This will further validate the utility of this metabolic BMI. The authors need to attempt to utilize the waist-circumference as an additional supportive association!

Response: Thank you for your valuable input. We, agree that incorporating markers for fat body distribution would complement our findings and provide a more comprehensive understanding of the association between metabolic BMI and different fat depots in the

body. We have examined the association of two additional measures of adiposity (waist circumference (WC) and waist to hip ratio (WHR)) with mBMI Δ . We observed that mBMI Δ had a stronger association with WC and WHR (Supp Figure 5) than with BMI itself (Figure 4D) suggesting that these measures may better capture the metabolic dysregulation associated with BMI. We have now highlighted this additional evidence in the manuscript (page 11, line 301-304).

- It would be great if the analysis for the metabolic score be developed based on common lipid markers such as LDL, TC, HDL, vLDL and TG to see how well these routine lipid markers fair with the calculated metabolic BMI score. The reason for this is to better understand if these commonly measured markers would be useful and comparable to predict the mBMI or not. A statement on this issue may be of value.

Response: Thank you for your suggestion. We, agree that incorporating common lipid markers (clinical lipids) into the analysis of the metabolic score would be valuable. We created models using just the clinical lipid (total cholesterol, HDL-C and triglycerides) along with age and sex and assessed how the resulting score aligns with the calculated metabolic BMI score using the lipidome. When we created models using the clinical lipid measures along with age and sex the model explained only 15.6 % variation in BMI in the AusDiab cohort. A statement on this is included in the manuscript (page 6, line 184-188) and (Supplementary Table 3, Supp Figure 3).

- This reviewer understands that the metabolomic profiling was done at baseline. However, it would enrich the paper if some samples from the same individuals were also measured longitudinally (at future time). Metabolomics change with time! Such exercise may reassure us that the observed mBMI was indeed maintained or changed especially in relation to the next point raised!!

Response: Thank you for your valuable feedback. We, appreciate your suggestion to enrich the findings using longitudinal metabolic measurements in the study if data were available at follow up. Although, examining metabolomic profiles over time would certainly provide insights into the stability or changes in mBMI within individuals overtime, unfortunately, we do not have longitudinal metabolite data. We are seeking funding to pursue these studies.

- The term metabolically healthy obese refers to a timepoint. The idea that almost all metabolically healthy obese convert with time to metabolically unhealthy obese need to be discussed in line with our work that needs to be cited and discussed (PMID: 33827711, PMID: 35109943).

Response: Thank you for your input. We acknowledge the significance of discussing the concept of metabolically healthy obesity (MHO) and its potential conversion to metabolically unhealthy obesity (MUO) over time. We have addressed this aspect in our manuscript on page 21 (line 580-583). Furthermore, we have briefly discussed the suggested studies in the introduction page 2, line 52-53 and lines 55-57) and discussion section (page 21 line 585-589).

-The authors mention that both cohorts are primarily of White/European ancestry. Can they add the exact numbers/percentages in Table 1? That would be easier for the reader than going back to previous publications. Would it be possible to adjust their analysis for ethnicity?

Response: Thank you for your valuable feedback and suggestion. Adding the exact numbers and percentages for the White/European ancestry in both cohorts to Table 1 (page 5) would indeed provide clearer information for readers. We have now updated the table by providing the percentage of ethnicity. While 100% of the BHS participants were white, in the Ausdiab 94.7% were white/European ancestry and 5.3% were Asian/other ancestry. While there was a significant difference in the BMI (higher for White/European ancestry compared to Asian) the mBMI Δ was only marginally higher in the Asia/other ethnicities (Supp Figure 12). However, the adjustment for ethnicity on top of age, sex, BMI and clinical lipids in our mBMI score-disease associations, didn't result in a significant change in the observed associations (data not shown). We have discussed this in the manuscript (page 21, line 589-592).

- Can the author discuss the main differences between the mBMI score that they are proposing and the metabolomic signatures proposed by others (e.g., Cirulli et al.2019) beyond explaining variation in BMI? Specifically, how sensitive is the mBMI compared to metabolomic signatures in predicting metabolic complications?

Response: the mBMI score devised in the current study is entirely based on lipidomic profile from a broad range of classes/subclasses; as such it represents a better coverage of lipid biology. However, metabolomic signatures proposed by others such as Cirulli et al 2019 and Ottosson et al 2022 are based on metabolites from different metabolic pathways although lipids constitute a proportion of these. Of note, in Cirulli's study lipids constitute only 47% of the measured metabolites out of their 650 full metabolites set and 40% lipids out of their 49 BMI associated metabolite set (page 20, lines 539-541); while in the Ottosson et al 2022, few metabolites (n=108) including acylcarnitine (as the only representative of lipids) were used to model BMI (ref 33) (page 20, lines 542-543). Consequently, the biological information captured by our score and previous scores is distinct although there will be a certain overlap of signal. We have discussed this in the manuscript (page 21, lines 547-556). Regarding the sensitivity of the score, while previous studies have attempted to capture the residual risk of cardiovascular outcomes particularly between the mBMI outlier groups, the signal being captured by metabolic BMI scores has not been clearly defined nor has the relationship with disease outcomes been adequately quantified (page 3, lines 99-101). We demonstrated the biological signal captured by mBMI score and precisely quantified the mBMI-disease associations by reporting odds ratios and hazard ratios and associated confidence intervals.

- Why is LDL-C not included in the analysis? Can the authors comment on that?

Response: We opted to exclude LDL-C from our analyses since it's a calculated measure. LDL-C is derived from total cholesterol and triglyceride levels using the Friedewald equation [ref 36]. Since LDL-C is already represented indirectly through its calculation based on these other lipid markers, including it as a separate variable in our analysis may introduce redundancy and multicollinearity issues. We have now mentioned this in the manuscript (page 6, line 188-189).

- How does waist circumference (as a measure of central obesity) associate with the newly- introduced score (mBMI and mBMI Δ)?

Response: We have looked at the association of two additional measures of adiposity (waist circumference (WC) and waist to hip ratio (WHR)) across the mBMI Δ groups/quintiles. Individuals with higher mBMI score such as those in the in Q5 of mBMI Δ ,

had a significantly higher WC and WHR than those in the Q1 ($p < 0.001$) (Supp Fig 5). Such a significant positive association of WC/WHR with the mBMI score, indicates that mBMI score effectively captures the degree of central obesity and its metabolic implications. We have now highlighted this supportive evidence in the manuscript (page 11, lines 301-304).

- Why was family history of diabetes adjusted for in the analysis for CVD risk (Table 2) but not in prevalent and incident T2DM risk (Supplementary Table 7)?

Response: Considering the familial history of diabetes and adjusting for it in our analysis is a methodologically sound approach. In response to this, we have revised the table to include the adjustment for family history of diabetes. However, it is worth noting that despite this adjustment, we did not observe significant changes or substantial modifications in our results (Supp Table 8 in the revised version).

- Using ridge regression model, the pBMI can explain 60.4% of the BMI variation in the AusDiab cohort and 40% of the BMI variation in the BHS cohort. Can the author suggest possible approaches to improve these numbers?

Response: In order to improve the explained variation of BMI, incorporating metabolites from diverse biochemical classes would be beneficial. A study by Cirulli et al. showed that increasing the number of metabolites from 49 to 650 resulted in an improvement from 43% to 49% in the explained BMI variation [ref 21] (discussion section, page 20, line 539-541). Additionally, adopting a multi-omics approach has shown to improve the prediction performances. For instance, the MetBMI model using 209 metabolites resulted in an explained variance of 68.9%, while the ProtBMI model based on proteomics data explained 70.6% of the variance. Combining metabolic, proteomic, and other chemical measurements further improved the R^2 , 0.78 [ref 34] (discussion section, page 20, line 544-547). However, rather than solely improving the prediction of BMI, we seek to gain insights into hidden metabolic risks by examining the differences between metabolome-defined BMI and actual BMI. Analysing this variance allows us to uncover valuable information about metabolic factors that are not adequately captured by BMI alone, facilitating a deeper understanding of underlying metabolic risks. In a scenario where nearly 100% of the BMI variance is explained, the score would be equivalent to BMI itself, precluding any additional metabolic health assessment beyond BMI.

-The authors should be more concise in their discussion on dietary and lifestyle factors. While they clearly show the association with mBMI, their title in the discussion “mBMI can be modified by dietary and lifestyle factors” gives the false impression that they actually conducted an intervention and showed that they can change mBMI by modifying diet or improving physical activity.

Response: The authors appreciate the observation regarding the discussion on dietary and lifestyle factors in relation to mBMI. Due to the journal's requirement, we have now removed all sub-headings in the discussion section. We have discussed the lifestyle-mBMI associations on pages 22-23, lines 641-662). Of note, it has been recently shown in an intervention study, in a subset of the Arivale study (a wellness program), that the mBMI can be effectively modified by diet or lifestyle [ref 34]. We have now updated the discussion including this reference (page 23, lines 657-662).

- Childhood obesity is currently classified using WHO growth charts after measuring BMI-for-age, can the authors add a paragraph in their discussion on whether their new score can be adapted for children and adolescents?

Thank you for your suggestion. The authors acknowledge your interest in exploring the potential adaptation of our newly developed score for children and adolescents in the classification of childhood obesity. It is important to note that the current score has been developed and validated specifically for use in adults. However, the authors recognize the significance of addressing the need for a specific population-specific score for children and adolescents in the future. We have included a brief paragraph about the potential adaptation of such as a score in youth in future (page 23 lines 671-674).

- The authors mention that compared to direct methods for measuring adiposity, BMI is an inexpensive, simple, and easily interpretable metric. Can they add a comparison in their discussion between these methods (e.g., computed tomography and dual energy x-ray absorptiometry) and performing targeted lipidomics from cost-effectiveness and clinical application?

Response: Thank you for your suggestion. The authors have now commented this in the discussion section “While direct measures of body fat distribution, such as computed tomography and dual-energy X-ray absorptiometry, have the potential to enhance risk assessment by providing valuable insights into body fat distribution, their practical application is constrained by high costs and inability to directly evaluate metabolic health and perturbations. In contrast, mBMI measures hold promise for understanding metabolic health and risk, albeit requiring development of their clinical utility and cost-effectiveness” (page 19, lines 494 - 499).

REVIEWERS' COMMENTS

Reviewer #1 (Remarks to the Author):

The revised manuscript has addressed all comments in a comprehensive manner. Overall, the study is of high interest, and demonstrates the value of clinical lipidomics. No additional revision needed.

Reviewer #2 (Remarks to the Author):

I am pleased to say that my concerns have been thoroughly addressed by the authors in a clear and methodical manner. The manuscript has been completed and will hopefully make a significant contribution to our understanding of metabolic BMI. I would like to extend my congratulations to the authors for this accomplishment.

Prof. Fahd Al-Mulla

CSO at the Dasman Diabetes Institute

Kuwait